

# Unfold: an integrated toolbox for overlap correction, non-linear modeling, and regression-based EEG analysis

Benedikt V. Ehinger[1,2] and Olaf Dimigen[3]

[1] Institute of Cognitive Science, Universität Osnabrück, Osnabrück, Germany
[2] Donders Institute for Brain, Cognition and Behaviour, Radboud University, Nijmegen, Netherlands
[3] Department of Psychology, Humboldt-Universität zu Berlin, Berlin, Germany

## ABSTRACT

Electrophysiological research with event-related brain potentials (ERPs) is increasingly moving from simple, strictly orthogonal stimulation paradigms towards more complex, quasi-experimental designs and naturalistic situations that involve fast, multisensory stimulation and complex motor behavior. As a result, electrophysiological responses from subsequent events often overlap with each other. In addition, the recorded neural activity is typically modulated by numerous covariates, which influence the measured responses in a linear or non-linear fashion. Examples of paradigms where systematic temporal overlap variations and low-level confounds between conditions cannot be avoided include combined electroencephalogram (EEG)/eye-tracking experiments during natural vision, fast multisensory stimulation experiments, and mobile brain/body imaging studies. However, even "traditional," highly controlled ERP datasets often contain a hidden mix of overlapping activity (e.g., from stimulus onsets, involuntary microsaccades, or button presses) and it is helpful or even necessary to disentangle these components for a correct interpretation of the results. In this paper, we introduce *unfold*, a powerful, yet easy-to-use MATLAB toolbox for regression-based EEG analyses that combines existing concepts of massive univariate modeling ("regression-ERPs"), linear deconvolution modeling, and non-linear modeling with the generalized additive model into one coherent and flexible analysis framework. The toolbox is modular, compatible with EEGLAB and can handle even large datasets efficiently. It also includes advanced options for regularization and the use of temporal basis functions (e.g., Fourier sets). We illustrate the advantages of this approach for simulated data as well as data from a standard face recognition experiment. In addition to traditional and non-conventional EEG/ERP designs, *unfold* can also be applied to other overlapping physiological signals, such as pupillary or electrodermal responses. It is available as open-source software at http://www.unfoldtoolbox.org.

Corresponding authors
Benedikt V. Ehinger,
behinger@uos.de
Olaf Dimigen,
olaf.dimigen@hu-berlin.de

## INTRODUCTION

Event-related brain responses in the electroencephalogram (EEG) are traditionally studied in strongly simplified and strictly orthogonal stimulus-response paradigms. In many cases, each experimental trial involves only a single, tightly controlled stimulation, and a single manual response. In recent years, however, there has been a rising interest in recording brain-electric activity also in more complex paradigms and naturalistic situations. Examples include laboratory studies with fast and concurrent streams of visual, auditory, and tactile stimuli (*Spitzer, Blankenburg & Summerfield, 2016*), experiments that combine EEG recordings with eye-tracking recordings during natural vision (*Dimigen et al., 2011*), EEG studies in virtual reality (*Ehinger et al., 2014*) or mobile brain/body imaging studies that investigate real-world interactions of freely moving participants (*Gramann et al., 2014*). There are two main problems in these types of situations: overlapping neural responses from subsequent events and complex influences of nuisance variables that cannot be fully controlled. However, even traditional event-related brain potential (ERP) experiments often contain a mixture of overlapping neural responses, for example, from stimulus onsets, involuntary microsaccades, or manual button presses.

Appropriate analysis of such datasets requires a paradigm shift away from simple averaging techniques towards more sophisticated, regression-based approaches (e.g., *Amsel, 2011*; *Pernet et al., 2011*; *Smith & Kutas, 2015b*; *Frömer, Maier & Abdel Rahman, 2018*; *Hauk et al., 2006*; *Van Humbeeck et al., 2018*) that can deconvolve overlapping potentials and also control or model the effects of both linear and non-linear covariates on the neural response. Importantly, the basic algorithms to deconvolve overlapping signals and to model the influences of both linear and non-linear covariates already exist. However, there is not yet a toolbox that integrates all of the necessary methods in one coherent workflow.

In the present paper, we introduce *unfold*, an open source, easy-to-use, and flexible MATLAB toolbox written to facilitate the use of advanced deconvolution models and spline regression in ERP research. It performs these calculations efficiently even for large models and datasets and allows to run complex models with a few lines of codes. The toolbox is programed in a modular fashion, meaning that intermediate analysis steps can be readily inspected and modified by the user if needed. It is also fully documented, can employ regularization, can model both linear and non-linear effects using spline regression, and is compatible with EEGLAB (*Delorme & Makeig, 2004*) a widely used toolbox to preprocess electrophysiological data that offers importers for many other biometric data formats, including eye-tracking and pupillometric data. *unfold* offers built-in functions to visualize the model coefficients (betas) of each predictor as waveforms or scalp topographies (i.e. "regression-ERPs," rERPs, *Burns et al., 2013*; *Smith & Kutas, 2015a*). Alternatively, results can be easily exported as plain text or transferred to other toolboxes like EEGLAB or Fieldtrip (*Oostenveld et al., 2011*). For statistical analyses at the group level, that is, second-level statistics, the resulting rERPs can be treated just like any other subject-level ERPs. As one suggestion, *unfold* integrates threshold-free cluster enhancement (TFCE) permutation tests for this purpose (*Smith & Nichols, 2009*; *Mensen & Khatami, 2013*).

In the following, we first briefly summarize some key concepts of regression-based EEG analysis, with an emphasis on linear deconvolution, spline regression, and temporal basis functions. We then describe the *unfold* toolbox that combines these concepts into one coherent framework. Finally, we illustrate its application to simulated data as well as real data from a standard ERP experiment. In particular, we will go through the typical steps to run and analyze a deconvolution model, using the data of a standard face recognition ERP experiment that contains overlapping potentials from three different sources: from stimulus onsets, from button presses, and from microsaccades, small eye movements that were involuntarily made by the participants during the task. We also give detailed descriptions of the features of the toolbox, including practical recommendations, simulation results, and advanced features such as regularization options or the use of temporal basis functions. We hope that our toolbox will both improve the understanding of traditional EEG datasets (e.g., by separating stimulus- and response-related components) as well as facilitate electrophysiological analyses in complex or (quasi-)natural situations, such as in combined eye-tracking/EEG and mobile brain/body imaging studies.

## A simple simulation example

Before we introduce a real dataset, let us first consider a simulated simple EEG/ERP study to illustrate the possibilities of the deconvolution approach. For this, let's imagine a typical stimulus-discrimination task with two conditions (Fig. 1): Participants are shown pictures of faces or houses and asked to classify the type of stimulus with a button press. Because this response is speeded, motor activity related to the preparation and execution of the manual response will overlap with the activity elicited by stimulus onset. Furthermore, we also assume that the mean reaction time (RTs) differs between the conditions, as it is the case in most experiments. In our example, if face pictures are on average classified faster than houses pictures (Fig. 1C), then a different overlap between stimulus- and response-related potentials will be observed in the two conditions. Importantly, as Fig. 1F shows, this will result in spurious conditions effects due to the varying temporal overlap alone, which can be easily mistaken for genuine differences in the brain's processing of houses and faces.

Human faces are also complex, high-dimensional stimuli with numerous properties that are difficult to perfectly control and orthogonalize in any given study. For simplicity, we assume here that the average luminance of the stimuli was not perfectly matched between conditions and is slightly, but systematically, higher for faces than houses (Fig. 1D). From previous studies, we know that the amplitude of the P1 visually-evoked potential increases as a non-linear (log) function of the luminance of the presented stimulus (*Halliday, 1982*), and thus we also simulate a logarithmic effect of luminance on the P1 of the stimulus-aligned ERP (Fig. 1E), which creates another spurious condition difference (Fig. 1G) in addition to that of varying response times (Fig. 1F).

Figures 1G and 1H show the same data modeled with *unfold*. Fortunately, with deconvolution modeling, we can not only remove the overlap effect (Fig. 1G), but by including luminance as a non-linear predictor, we simultaneously also control the influence of this covariate (Fig. 1H). How this is done is explained in more detail in the following.

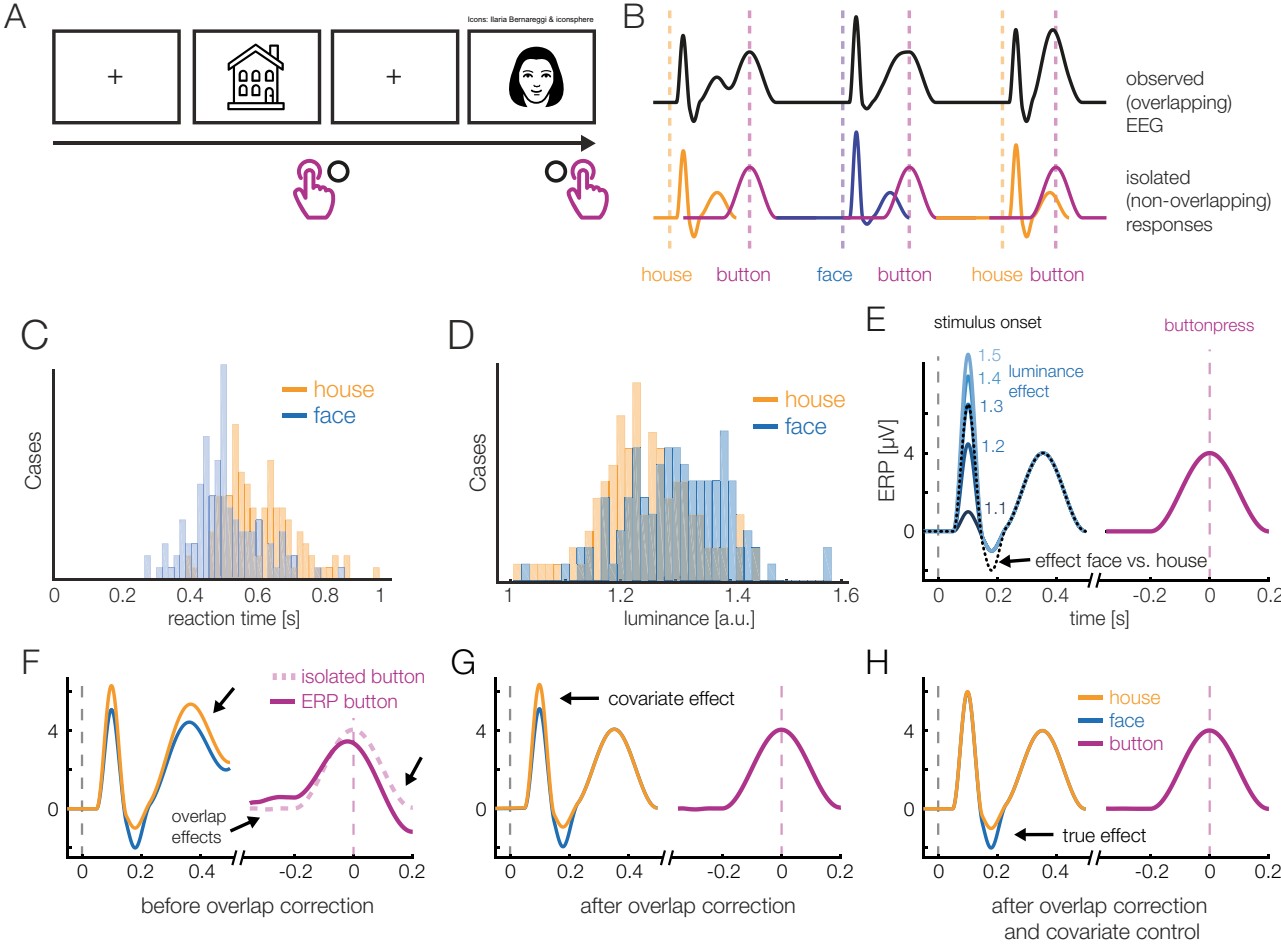

**Figure 1 A hypothetical simple ERP experiment with overlapping responses and a non-linear covariate.** (A) A hypothetical simple ERP experiment with overlapping responses and a non-linear covariate. Data in this figure was simulated and then modeled with *unfold*. Participants saw pictures of faces or house and categorized them with a button press. (B) A short interval of the recorded EEG. Every stimulus onset and every button press elicits a brain response (isolated responses). However, because brain responses to the stimulus overlap with that to the response, we can only observe the sum of the overlapping responses in the EEG (upper row). (C) Because humans are experts for faces, we assume here that they reacted faster to faces than houses, meaning that the overlap with the preceding stimulus-onset ERP is larger in the face than house condition. (D) Furthermore, we assume that faces and house stimuli were not perfectly matched in terms of all other stimulus properties (e.g., spectrum, size, shape). For this example, let us simply assume that they differed in mean luminance. (E) The N170 component of the ERP is typically larger for faces than houses. In addition, however, the higher luminance alone increases the amplitude of the visual P1 component of the ERP. Because luminance is slightly higher for faces and houses, this will result in a spurious condition difference. (F) Average ERP for faces and houses, without deconvolution modeling. In addition to the genuine N170 effect (larger N170 for faces), we can see various spurious differences, caused by overlapping responses and the luminance difference. (G) Linear deconvolution corrects for the effects of overlapping potentials. (H) To also remove the confounding luminance effect, we need to also include this predictor in the model. Now we are able to only recover the true N170 effect without confounds (a similar figure was used in *Dimigen & Ehinger, 2019*).

## Existing deconvolution approaches

Deconvolution methods for EEG have existed for some time (*Hansen, 1983*; *Eysholdt & Schreiner, 1982*), but most older deconvolution approaches show severe limitations in their applicability. They are either restricted to just two different events (*Hansen, 1983*; *Zhang, 1998*), require special stimulus sequences (*Eysholdt & Schreiner, 1982*; *Marsh, 1992*; *Delgado & Ozdamar, 2004*; *Jewett et al., 2004*; *Wang et al., 2006*), rely on semi-automatic, iterative methods like ADJAR (*Woldorff, 1993*) that can be slow or difficult to

converge (*Talsma & Woldorff, 2004*; *Kristensen, Rivet & Guérin-Dugué, 2017b*), or were tailored for special applications. In particular, the specialized RIDE algorithm (*Ouyang et al., 2011*; *Ouyang, Sommer & Zhou, 2015*) offers a unique feature in that it is able to deconvolve time-jittered ERP components even in the absence of a designated event marker. However, while RIDE has been successfully used to separate stimulus- and response-related ERP components (*Ouyang et al., 2011*; *Ouyang, Sommer & Zhou, 2015*), it does not support continuous predictors and is intended for a small number of overlapping events.

In recent years, an alternative deconvolution method based on the linear model has been proposed and successfully applied to the overlap problem (*Lütkenhöner, 2010*; *Dandekar et al., 2012a*; *Litvak et al., 2013*; *Spitzer, Blankenburg & Summerfield, 2016*; *Kristensen, Rivet & Guérin-Dugué, 2017a*, *2017b*; *Sassenhagen, 2018*; *Cornelissen, Sassenhagen & Võ, 2019*; *Coco, Nuthmann & Dimigen, 2018*; *Bigdely-Shamlo et al., 2018*; *Dimigen & Ehinger, 2019*). This deconvolution approach was first applied extensively to fMRI data (*Dale & Buckner, 1997*) where the slowly varying BOLD signal overlaps between subsequent events. However, in fMRI, the shape of the BOLD response is well-known and this prior knowledge allows the researcher to use model-based deconvolution. If no assumptions about the response shape (i.e., the kernel) are made, the approach used in fMRI is closely related to the basic linear deconvolution approach discussed below.

## Deconvolution within the linear model

With deconvolution techniques, overlapping EEG activity is understood as the linear convolution of experimental event latencies with isolated neural responses (Fig. 1B). The inverse operation is deconvolution, which recovers the unknown isolated neural responses given only the measured (convolved) EEG and the latencies of the experimental events (Fig. 1H). Deconvolution is possible if the subsequent events in an experiment occur with varying temporal overlap, in a varying temporal sequence, or both. In classical experiments, stimulus-onset asynchronies and stimulus sequences can be varied experimentally and the latencies of motor actions (such as saccades or button presses) also vary naturally. This varying overlap allows for modeling of the unknown isolated responses, assuming that the overlapping signals add up linearly. More specifically, we assume (1) that the electrical fields generated by the brain sum linearly (a justified assumption, see *Nunez & Srinivasan, 2006*) and (2) that the overlap, or interval between events, does not influence the computations occurring in the brain—and therefore the underlying waveforms (see also Discussion).

The benefits of this approach are numerous: The experimental design is not restricted to special stimulus sequences, multiple regression allows modeling of an arbitrary number of different events, and the mathematical properties of the linear model are well understood.

## Linear deconvolution

The classic massive univariate linear model, *without* overlap correction, is applied to epoched EEG data and can be written as:

$$\mu_{i,\tau} = X_i \beta_\tau \text{ with } y_{i,\tau} \sim \text{normal}(\mu_{i,\tau}, \sigma_\tau)$$

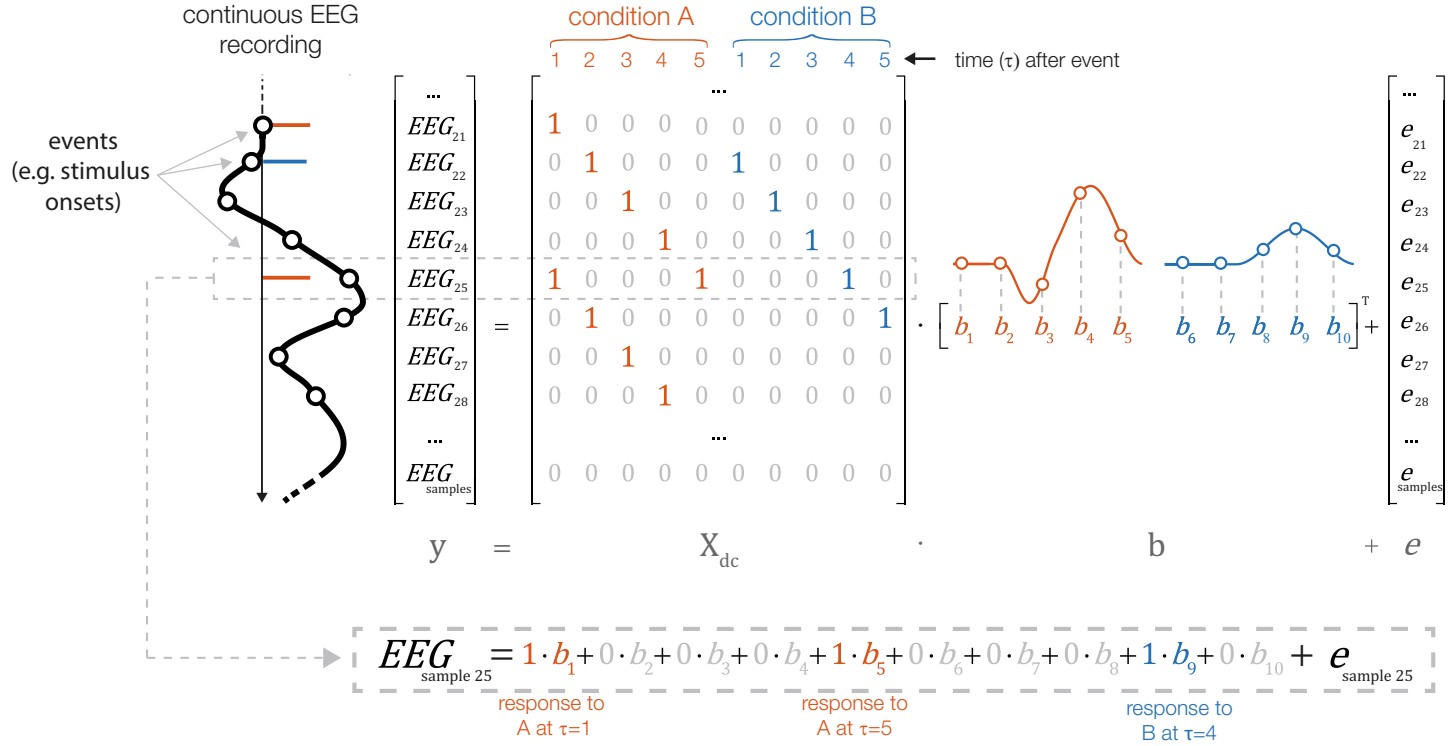

**Figure 2 Linear deconvolution by time expansion.** Linear deconvolution explains the continuous (toy) EEG signal within a single regression model. Specifically, we want to estimate the response (betas) evoked by each event so that together, they best explain the observed EEG. For this purpose, we create a time-expanded version of the design matrix ($X_{dc}$) in which a number of time points around each event (here: only 5 points) are added as predictors. We then solve the model for b, the betas. For instance, in the model above, sample number 25 of the continuous EEG recording can be explained by the sum of responses to three different experimental events: the response to a first event of type "A" (at time point 5 after that event), by the response to an event of type "B" (at time 4 after that event) and by a second occurrence of an event of type "A" (at time 1 after that event). Because the sequences of events and their temporal distances vary throughout the experiment, it is possible to find a unique solution for the betas that best explains the measured EEG signal. These betas, or "regression-ERPs" can then be plotted and analyzed like conventional ERP waveforms. Figure adapted from *Dimigen & Ehinger (2019)*, with permission).

Here, $X$ is the design matrix. It has $i$ rows (each describing one instance of an event of type e) and c columns (each describing the status of one predictor).

Furthermore, let $\tau$ be the "local time" relative to onset of the event (e.g., −100 to +500 sampling points). $\mu_{i,\tau}$ is the expected (average) EEG signal measured after event $i$ that we wish to predict at a given time point $\tau$ relative to the event onset. $\beta$ is a vector of unknown parameters that we wish to estimate for each time point in the epoched EEG data. Importantly, therefore, this approach fits a separate linear model at each time point $\tau$. A single entry in the design matrix will be referred by lowercase $x_{i,c}$.

In contrast, with linear deconvolution we enter the continuous EEG data into the model. We then make use of the knowledge that each observed sample of the continuous EEG can be described as the linear sum of (possibly) several overlapping event-related EEG responses. Depending on the latencies of the neighboring events, these overlapping responses occur at different times $\tau$ relative to the current event (see Fig. 2). That is, in the example in Fig. 2, where the responses of two types of events, A and B, overlap with each other, the observed continuous EEG at time point $t$ of the continuous EEG recording can

be described as follows:

$$\text{EEG}_{t=25} = 1\beta_{A,1} + 0\beta_{A,2} + 0\beta_{A,3} + 0\beta_{A,4} + 1\beta_{A,5} + 0\beta_{B,1} + 0\beta_{B,2} + 0\beta_{B,3} + 1\beta_{B,4} + 0\beta_{B,5}$$

In the example in Fig. 2, the spontaneous EEG at time-point $t$ is modeled as the linear sum of a to-be-estimated response to the first instance of event type A at local time $\tau = 5$ (i.e., from the point of view of EEG($t$) this instance occurred 5 time samples before), another response to the second instance of event type A at local time $\tau = 0$ (i.e., this instance just occurred at $t$), and another response to the instance of event type B at local time $\tau = 4$.

The necessary design matrix to implement this model, $X_{dc}$, will span the duration of the entire EEG recording. It can be generated from any design matrix $X$ by an algorithm we will call *time expansion* in the following. In this process, each predictor in the original design matrix will be expanded to several columns, which then code a number of "local" time points relative to the event onset. An example for a time-expanded design matrix is shown in Fig. 2.

## Time expansion

The process to create the time-expanded design matrix $X_{dc}$ is illustrated in Fig. 2. In the following sections, we will describe the construction of $X_{dc}$ more formally.

Let $t$ be the time of the continuous EEG signal $y$, which keeps increasing throughout the experiment. $\tau$ is still the local time, that is, the temporal distance of an EEG sample relative to an instance of event $e$. Let $i$ be the instance of one such event. $X_i$ is therefore the accompanying row of the design matrix $X$ which specifies the predictors for each event of type $e$. The design matrix X consists of multiple columns $c$, each representing one predictor (for which we want to estimate the accompanying $\beta$).

$X_{dc}$ can be constructed from multiple concatenated, time-expanded square diagonal matrices $G$ with size $\tau$ one for each instance of the event $e$. For the purpose of illustration, it is helpful to construct the design matrix first for just a single predictor and a single instance of a single type of event (e.g., a manual response). Afterwards, we will add multiple predictors, then multiple instances of a single event type and finally multiple different event types (e.g., stimuli and responses).

### Single predictor, single instance, single event type

The matrix $G_c$ for a single predictor, single type of event, and single instance of this event type is square diagonal where the size is specified by the number of samples around the event instance onsets to be taken into account:

$$G_c = I x_{i,c} = \begin{bmatrix} x_{i,c} & 0 & 0 & 0 \\ 0 & x_{i,c} & 0 & 0 \\ 0 & 0 & x_{i,c} & 0 \\ 0 & 0 & 0 & x_{i,c} \end{bmatrix}$$

It is a scaling of an identity matrix by the scalar $x_{i,c}$ which is a single entry of the design matrix $X$ defining the predictor $c$ at the single instance $i$. In the case of a dummy-coded variable (0 or 1) we would get either a matrix full of zeros or the identity matrix; in the case of a continuous predictor we get a scalar matrix where the diagonal of $G_{i,c}$ contains the continuous predictor value.

### Multiple predictors, single instance, single event type

In the case of multiple predictors $c$, we generate multiple matrices $G_{i,c}$ and concatenate them to $G^* = [G_1 \ldots G_c]$. Therefore, a matrix with two predictors at the instance $i$ of an event $e$ could look like this:

$$G^* = [G_1 G_2] = \begin{bmatrix} 1 & 0 & 0 & 0 & 10 & 0 & 0 & 0 \\ 0 & 1 & 0 & 0 & 0 & 10 & 0 & 0 \\ 0 & 0 & 1 & 0 & 0 & 0 & 10 & 0 \\ 0 & 0 & 0 & 1 & 0 & 0 & 0 & 10 \end{bmatrix}$$

### Multiple predictors, multiple instances, single events type

In the case of multiple instances of the same event, we have one $G^*$ matrix for every instance. We combine them into a large matrix $X_{dc}$ (where dc stands for deconvolution) by inserting the $G^*$ matrices into $X_{dc}$ around the time points (in continuous EEG time $t$) where the instance of the event occurred. Because $\tau$ (and therefore $G^*$) is usually larger than the time distance between two event instances, we insert rows of multiple $G^*$ matrices in an overlapping (summed) way. Consequently, we model the same time point of the EEG by the combined rows of multiple $G^*$ matrices (Figs. 2 and 3A). By solving the linear system with $X_{dc}\beta$ for $\beta$ we effectively deconvolve the original signal.

### Multiple predictors, multiple instances, multiple events types

We usually have multiple different types of events $e_1, e_2, \ldots$ For each of these event types, we create one $X_{dc}^e$ matrix as described above. Each $X_{dc}^e$ matrix spans $t$ rows and thus, the continuous EEG signal. To get the final matrix $X_{dc}$ we simply concatenate them along the columns before the model inversion.

$$X_{dc} = \begin{bmatrix} X_{dc}^1 \ldots X_{dc}^e \end{bmatrix}$$

Similarly, if we wanted to include a continuous covariate spanning the whole duration of the continuous EEG signal (see Discussion), for example some feature of a continuous audio signal (Crosse et al., 2016), we could simply concatenate it as an additional column to the design matrix.

The formula for the deconvolution model is then:

$$\mu_t = X_{dc,t}\beta \text{ with } y_t \sim normal(\mu_t, \sigma)$$

$\mu_t$ is the expected value of the continuous EEG signal $y_t$. This time-expanded linear model simultaneously fits all the parameters $\beta$ describing the deconvolved rERPs of interest. This comes at the cost of a very large size (*number of continuous EEG samples × number of predictor columns*). Fortunately, this matrix is also very sparse (containing mostly zeros) and can therefore be efficiently solved with modern sparse solvers. For further detail see the excellent tutorial reviews by Smith & Kutas (2015a, 2015b).

## Modeling non-linear effects with spline regression

Spline regression is a method to estimate non-linear relationships between a continuous predictor and an outcome. In the simple case of a single predictor it can be understood as
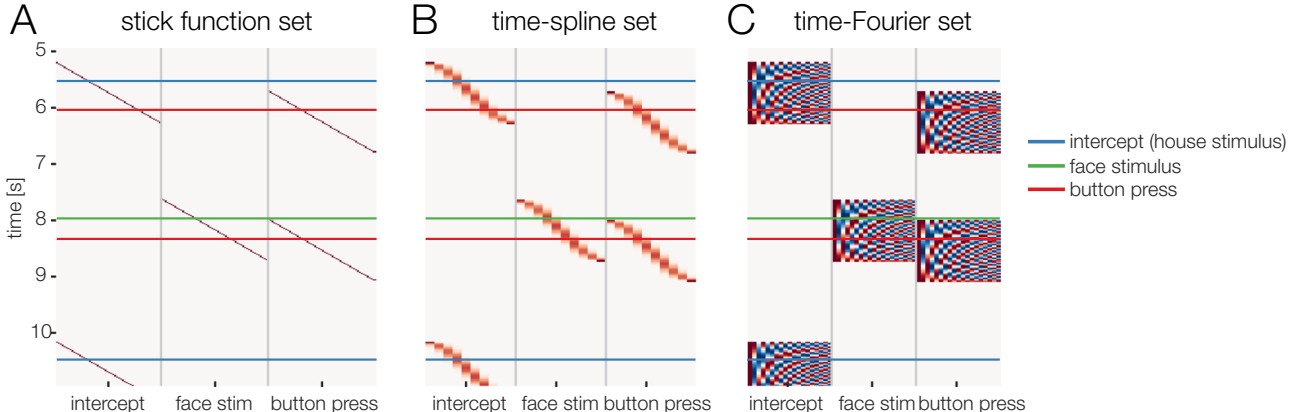

**Figure 3 Temporal basis functions.** Overview over different temporal basis functions. The expanded design matrix $X_{dc}$ is plotted, the $y$-axis represents time and the $x$-axis shows all time-expanded predictors in the model. In *unfold*, three methods are available for time expansion: (A) Stick-functions. Here, each modeled time point relative to the event is represented by a unique predictor. (B) Time-splines allow neighboring time points to smooth themselves. This generally results in less predictors than the stick function set. (C) Truncated time-Fourier set: It is also possible to use a Fourier basis. By omitting high frequencies from the Fourier-set, the data are effectively low-pass filtered during the deconvolution process (see also Fig. 6). <br>

a type of local smoother function. For a detailed introduction to spline regression we recommend *Harrell (2015)* or *Wood (2017)*. We will now outline how we can use spline predictors to model non-linear effects within the linear regression framework. We follow the definition of a generalized additive model (GAM) by *Wood (2017*, pp. 249–250*)*:

$$\mu_i = X_i\beta + \sum_j z_j f_j(x_i)$$

The sum $\sum_j z_j f_j(x_i)$ represents a basis set with $j$ unknown parameters $z$ (analog to the $\beta$ vector). The time-indices were omitted here. The most common example for such a basis set would be polynomial expansion. Using the polynomial basis set with order three would results in the following function:

$$\mu_i = X_i\beta + z_1 x^1 + z_2 x^2 + z_3 x^3$$

However, due to several suboptimalities of the polynomial basis (*Runge, 1901*) we will make use of the cubic B-spline basis set instead. Spline regression is conceptually related to the better-known polynomial expansion, but instead of using polynomials, one uses locally bounded functions. In other words, whereas a polynomial ranges over the whole range of the continuous predictor, a B-spline is restricted to a local range.

This basis set is constructed using the De-Casteljau algorithm (*De Casteljau, 1959*) implemented by *Luong (2016)*. It is a basis set that is strictly local: Each basis function is non-zero only over a maximum of 5 other basis functions (for cubic splines; see *Wood, 2017*, p. 204*)*.

Multiple terms can be concatenated, resulting in a GAM:

$$\mu_i = X_i\beta + \sum_j z_j f_j(x_{1i}) + \sum_j z_j f_j(x_{2i})$$

If interactions between two non-linear spline predictors are expected, we can also make use of two-dimensional splines:

$$\mu_i = X_i\beta + \sum_j z_j f_j(x_{1i}, x_{2i})$$

In the *unfold* toolbox, 2D-splines are created based on the pairwise products between $z_{1,j}$ and $z_{2,j}$. Thus, a 2D spline between two spline-predictors with 10 spline-functions each would result in 100 parameters to be estimated.

The number of basis functions to use for each non-linear spline term is usually determined by either regularization or cross-validation. Cross-validation is difficult in the case of large EEG datasets (see also Discussion) and we therefore recommend to the number of splines (and thus the flexibility of the non-linear predictors) prior to the analysis.

## Using temporal basis functions

In the previous section, we made use of time-expansion to model the overlap. For this, we used the so-called stick function approach (also referred to as FIR or dummy-coding). That is, each time-point relative to an event (local time) was coded using a 1 or 0 (in the case of dummy-coded variables), resulting in staircase patterns in the design matrix (cf. Figs. 2 and 3A). However, this approach is computationally expensive. Due to the high sampling rate of EEG (typically 200–1,000 Hz), already a single second of estimated ERP response requires us to estimate 200–1,000 coefficients per predictor. Therefore, some groups started to use other time basis sets to effectively smooth the estimated rERPs (*Litvak et al., 2013*; but see *Smith & Kutas, 2015b*).

We will discuss two examples here: The time-Fourier set (*Litvak et al., 2013*; *Spitzer, Blankenburg & Summerfield, 2016*) and—newly introduced in this paper—the time-spline set. In the time-spline set, adjacent local time coefficients are effectively combined according to a spline set (Fig. 3B). Splines are a suitable basis function because EEG signals are smooth and values close in time have similar values. The number of splines chosen here defines the amount of smoothing of the resulting deconvolved ERP. The same principle holds for the time-Fourier set. Here, we replace the stick-function set with a truncated Fourier set (Fig. 3C). Truncating the Fourier set at high frequencies effectively removes high frequencies from the modeled ERP and can therefore be thought of as a low-pass filter. A benefit of using (truncated) temporal basis functions rather than the simple stick functions is that fewer unknown parameters need to be estimated, at the cost of temporal precision. It is therefore possible that this results in numerically more stable solutions to the linear problem, because we are constraining the solution space. Low-pass filtering and downsampling of the data followed by using the stick function deconvolution might result in better control of the spectral properties.

## Existing toolboxes

To our knowledge, no existing toolbox supports non-linear, spline-based general additive modeling of EEG activity. Also, we were missing a toolbox that solely focuses on deconvolving signals and allowed for a simple specification of the model to be fitted, for example, using the commonly used Wilkinson notation (as also used, e.g., in *R*).

A few other existing EEG toolboxes allow for deconvolution analyses, but we found that each has their limitations. Plain linear modeling (including second-level analyses) can be performed using the LIMO toolbox (*Pernet et al., 2011*), but this toolbox does not support deconvolution analyses or spline regression. To our knowledge, five toolboxes support some form of deconvolution: SPM (*Litvak et al., 2013*; *Penny et al., 2006*), the rERP extension for EEGLAB (*Burns et al., 2013*), pyrERP (*Smith, 2013*), mTRF (*Crosse et al., 2016*), and MNE (*Gramfort et al., 2014*). SPM allows for deconvolution of linear responses using Fourier temporal basis sets. However, in order to make use of these deconvolution functions, quite a bit of manual coding is needed. The rERP extension for EEGLAB and the pyrERP toolbox for Python both allow for estimation of linear models and deconvolution; however, both toolboxes appear not to be maintained anymore; rERP is currently non-functional (for current MATLAB versions) and no documentation is available for pyERP. The MNE toolbox is a general-purpose Python-based EEG processing toolbox that supports both deconvolution and massive univariate modeling. It is actively maintained and some basic tutorials are available. The mTRF toolbox is a special type of deconvolution toolbox designed to be used with continuous predictors (e.g., auditory streams) that last over the whole continuous EEG recording (see Discussion).

## THE UNFOLD TOOLBOX

In the following, we describe basic and advanced features available in the *unfold* toolbox and also give practical recommendations for problems that researchers might experience in the modeling process. Specifically, we describe how to (1) specify the model via Wilkinson formulas, (2) include non-linear predictors via spline regression, (3) model the data with basis functions over time (e.g., a Fourier basis set), (4) impute missing data in the design matrix, (5) treat intervals of the continuous EEG containing EEG artifacts (e.g., from muscle activity or skin conductance changes), (5) specify alternative solvers (with regularization) that can solve even large models in a reasonable time, and (6) run the same regression model both as a deconvolution model and also a mass multivariate model without deconvolution. Finally, we summarize options for (7) visualizing and (8) exporting the results (Fig. 4).

### Data import

As a start, we need a data structure in EEGLAB format (*Delorme & Makeig, 2004*) that contains the continuous EEG data and event codes. In traditional EEG experiments, events will typically be stimulus and response triggers, but many other types of events are also possible (e.g., voice onsets, the on- or offsets of eye or body movements etc.). In most cases, the EEG data entered into the model should have already been corrected for biological and technical artifacts (e.g., ocular artifacts, scalp muscle EMG, or power line noise), for example, with independent component analysis (ICA).

### Specifying models using Wilkinson notation

We begin the modeling process by specifying the model formula and by generating the corresponding design matrix *X*. In the *unfold* toolbox, models are specified using the intuitive Wilkinson notation (*Wilkinson & Rogers, 1973*) also commonly used in *R*, the

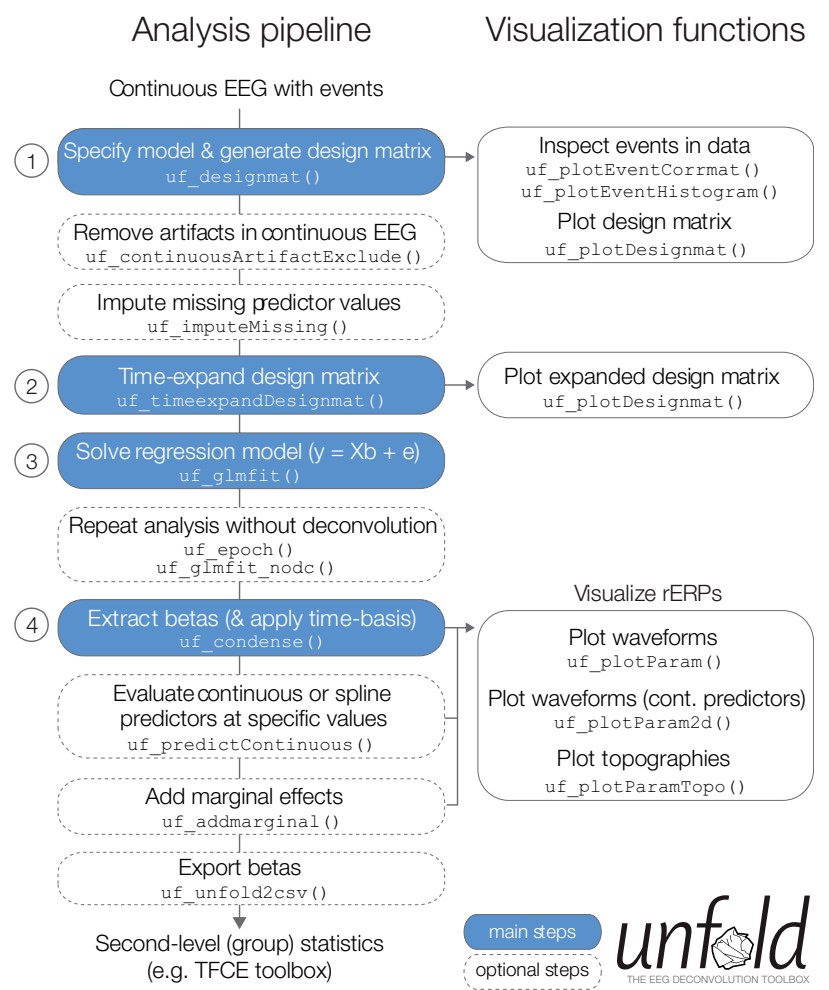

**Figure 4 Overview over typical analysis steps with *unfold*.** The first step is to load a continuous EEG dataset into EEGLAB. This dataset should already contain event markers (e.g., for stimulus onsets, button presses, etc.). Afterwards there are four main analysis steps, that can be executed with a few lines of code (see also Box 1). These steps, highlighted in blue, are: (1) Define the model formula and let *unfold* generate the design matrix, (2) time-expand this design matrix, (3) solve the model to obtain the betas (i.e., rERPs), and (4) convert the betas into a convenient format for plotting and statistics. The right column lists several inbuilt plotting functions to visualize intermediate analysis steps or to plot the results (see also Fig. 8).

Matlab statistics toolbox, and python StatsModels. For example, for the hypothetical face/house experiment depicted in Fig. 1, we might define the following model:

$$\texttt{EEG} \sim \texttt{1 + cat(is\_face) + luminance}$$

More generally, we can also specify more complex formulas, such as:

$$\texttt{EEG} \sim \texttt{1 + cat(predictor1) + predictor2 + spl(predictor3, 5)}$$

Here, `cat()` specifies that the `predictor1` should be dummy encoded as a categorical variable or factor rather than treated as a continuous variable. If a variable is already dummy-coded as 0/1 it is not strictly necessary to add the `cat()` command, but it would

be necessary to specify multi-level categorical variables (3 levels or more). In contrast, `predictor2` should be modeled as continuous linear covariate and `predictor3` as a non-linear spline predictor. In the formula, a plus sign (+) means that only the main effects will be modeled. Interactions between predictors are added by replacing the + with a * or a :, depending on whether all main effects and interactions should be modeled (*), or only the interactions (:). In *unfold*, the type of coding (dummy/treatment/reference or effect/ contrast/sum coding) can be selected. If the default treatment coding is used, the predictors will represent the difference to the intercept term (coded by the 1). The reference level of the categorical variable and the ordering of the levels is determined alphabetically or can be specified by the user manually. The toolbox also allows to specify different formulas for different events. For example, stimulus onset events can have a different (e.g., more complex) formula than manual response events.

Once the formula is defined, the design matrix $X$ is time-expanded to $X_{dc}$ and now spans the duration of the entire EEG recording. Subsequently for each channel, the equation (EEG = $X_{dc}$ * $b$ + $e$) is solved for "$b$", the betas, which correspond to subject-level rERP waveforms. For example, in the model above, for which we used treatment coding, the intercept term would correspond to the group-average ERP. The other betas, such as those for `cat(is_face)`, will capture the partial effect of that particular predictor, corresponding to a difference wave in traditional ERPs (here: face-ERP minus house-ERP).

In the same linear model, we can simultaneously model brain responses evoked by other experimental events, such as button presses. Each of these other event types can be modeled by its own formula. In our face/house example, we would want to model the response-related ERP that is elicited by the button press at the end of the trial, because this ERP will otherwise overlap to a varying degree with the stimulus-ERP. We do this by defining an additional simple intercept model for all button press events. In this way, the ERP evoked by button presses will be removed from the estimation of the stimulus ERPs. The complete model would then be:

$$\text{EEG}_{\text{fix}} \sim 1 + \texttt{cat(is\_face)} + \texttt{spl(luminance, 5)} \quad \{\text{for stimulus onset events}\}$$
$$\text{EEG}_{\text{button}} \sim 1 \quad \{\text{for manual button press events}\}$$

### Spline regression to model (non-linear) predictors

As explained earlier, many influences on the EEG are not strictly linear. In addition to linear terms, one can therefore use cubic B-splines to perform spline regression, an approach commonly summarized under the name GAM. An illustration of this approach is provided in Fig. 5. In the *unfold* toolbox, spline regression can be performed by adding `spl()` around the predictor name, as for `predictor3` in the formula above, which specifies a model using 5 B-splines instead of a continuous linear predictor. We can model covariates as non-linear predictors:

$$\text{EEG} \sim 1 + \texttt{spl (A, 5)} + \texttt{2dspl (B, C, 5)} + \texttt{circspl (D, 5, 0, 360)}$$

With this formula, the effect "A" would be modeled by a basis set consisting of five splines. We would also fit a 2D spline between continuous variable B and C with five

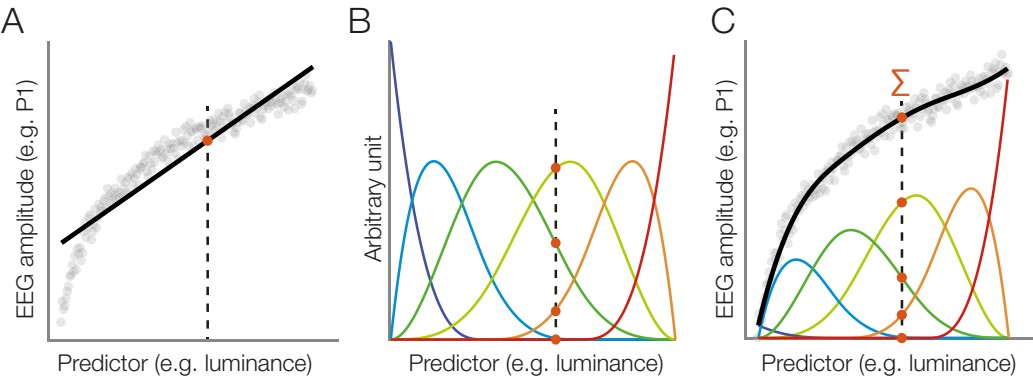

**Figure 5 Modeling a non-linear relationship with a set of spline functions.** (A) Example of a non-linear relationship between a predictor (e.g., stimulus luminance) and a dependent variable (e.g., EEG amplitude). A linear function (black line) does not fit the data well. We will follow one luminance value (dashed line) at which the linear function is evaluated (red dot). (B) Instead of a linear fit, we define a set of overlapping spline functions which are distributed across the range of the predictor. In this example, we are using a set of six b-splines. For our luminance value, we receive six new predictor values. Only three of them are non-zero. (C) We weight each spline with its respective estimated beta value. To predict the dependent variable (EEG amplitude) at our luminance value (dashed line), we sum up the weighted spline functions (red dots). Because the splines are overlapping, this produces a smooth, non-linear fit to the observed data.

splines each. In addition, we would fit a circular spline based on covariate D using five splines with the limits 0° and 360° being the wrapping point of the circular spline.

In *unfold*, three spline functions are already implemented. For B-splines we use the de Casteljau algorithm implemented by Bruno Luong. For interactions between spline-modeled covariates, first the default spline function is used on each predictor to generate $n$ splines. Then the resulting vectors are elementwise multiplied with each other, generating $n^2$ final predictors. For cyclical predictors such as the angle of a saccadic eye movement (which ranges, e.g., from 0 to 2π), it is possibly to use cyclical B-splines, as explained above. These are implemented based on code from *patsy*, a python statistical package (https://patsy.readthedocs.io) which follows an algorithm described in *Wood (2017*, pp. 201–205*)*. For maximal flexibility, we also allow the user to define custom spline functions. This would also allow to implement other basis sets, for example, polynomial expansion.

In our default B-spline implementation, the so-called *knots* define the peaks of the cubic splines, except for the boundaries (i.e., extremes) of the covariate (see Fig. 5B). The resulting number of splines are given by the number of knots minus 4 (4 due to the cubic nature, see *Wood, 2017*, p. 204), but the user only has to specify the number of basis functions, not the exact number of knots. The placement of knots (and therefore the number of splines) are critical parameters to appropriately model the predictor and to avoid over- or underfitting the data. The toolbox's default knot placement is on the quantiles of the predictor, which will increase resolution of the splines in areas where there are a lot of data points and offers stronger smoothing in other areas, where predictor values are sparser (similar to *Harrell, 2015*, p. 26). This can be changed by users who want to use a custom sequence of knots. Generalized cross-validation or penalized regression

could be used to narrow down the number of knots to be used but is computational expensive and currently not supported in *unfold*.

## Using time basis functions

Temporal basis functions were introduced earlier. The stick-function approach, as also illustrated in Figs. 2 and 3A, is the default option in *unfold*. As alternatives, it is also possible to employ either a Fourier basis set or a set of *temporal* spline function. For example, for the time-expansion step, *Litvak et al. (2013*; *Spitzer, Blankenburg & Summerfield, 2016)* used a Fourier basis sets instead of stick-functions. Figure 6 compares simulation results for stick functions with those obtained with a Fourier basis set and a spline basis set in terms of the spectral components and the resulting filter artifacts. At this point, more simulation studies are needed to understand the effects of temporal basis sets on EEG data. We therefore follow the recommendation of *Smith & Kutas (2015b)* to use stick-functions for now.

## Imputation of missing values

If a predictor has a missing value in massive univariate regression models, it is typically necessary to remove the whole trial. One workaround for this practical problem is to impute (i.e., interpolate) the value of missing predictors. In the deconvolution case, imputation is even more important for a reliable model fit, because if a whole event is removed, then overlapping activity from this event with that of the neighboring events would not be accounted for. In *unfold* we therefore offer several algorithms to treat missing values: the dropping of events with missing information, or imputation by the marginal, mean, or median values of the other events.

## Dealing with EEG artifacts

Linear deconvolution needs to be performed on continuous, rather than epoched data. This creates challenges with regard to the treatment of intervals that contain EEG artifacts. The way to handle artifacts in a linear deconvolution model is therefore to detect—but not to remove—the intervals containing artifacts in the continuous dataset. For these contaminated intervals, the time-expanded design matrix ($X_{dc}$) is then blanked out, that is, filled with zeros, so that the artifacts do not affect the model estimation (*Smith & Kutas, 2015b*). If the data would have been cleaned prior to the time-expansion step, then events might have been removed that would overlap with clean data segments.

Of course, this requires the researcher to use methods for artifact correction that can be applied to continuous rather than segmented data (such as ICA). Similarly, we need methods that can detect residual artifacts in the continuous rather than epoched EEG. One example would be a peak-to-peak voltage threshold that is applied within a moving time window (shifted step-by-step across the entire recording). Whenever the peak-to-peak voltage within the window exceeds a given threshold, the corresponding interval would then be blanked out in the design matrix. Detecting artifacts in the continuous rather than segmented EEG also has some small additional benefit, because if the data of a trial is only partially contaminated, the clean parts can still enter the model estimation (*Smith & Kutas, 2015b*).

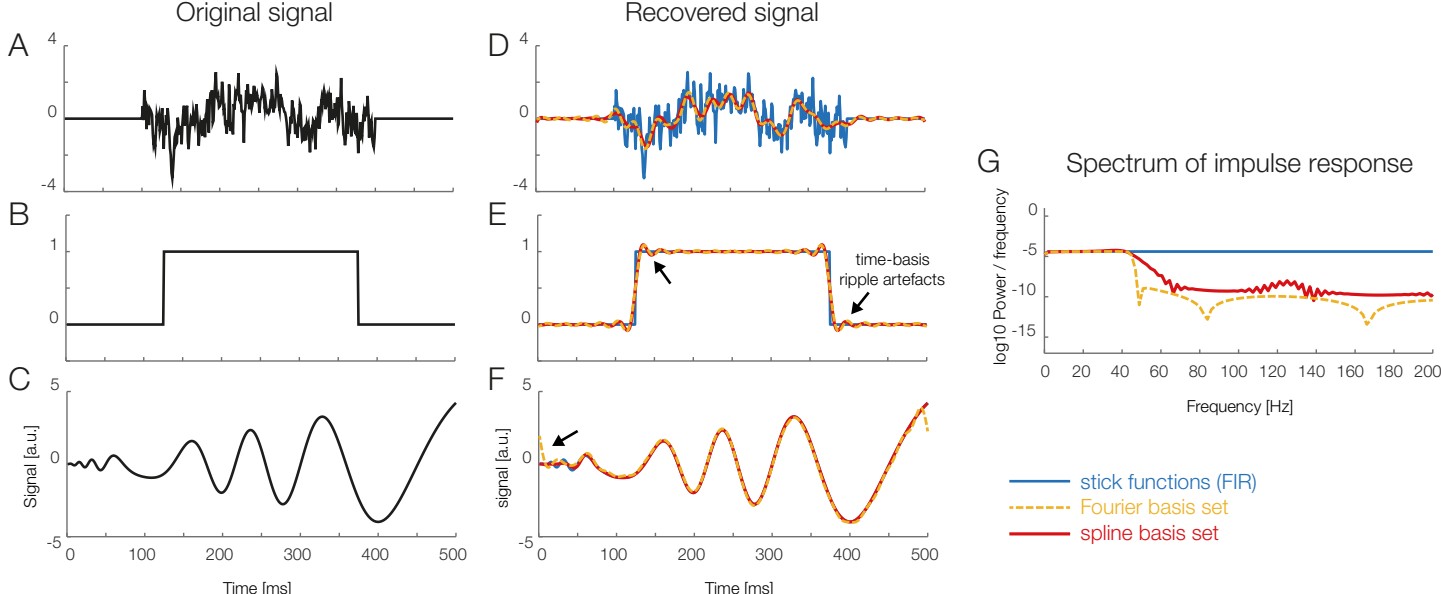

**Figure 6 Using temporal basis functions.** Effect of using different time basis functions on the recovery of the original signal using deconvolution. (A–C) Show three different example signals without deconvolution (in black) and with convolution using different methods for the time-expansion (stick, Fourier, spline). We zero-padded the original signal to be able to show boundary artifacts. For the analysis we used 45 time-splines and in order to keep the number of parameters equivalent, the first 22 cosine and sine functions of the Fourier set. The smoothing effects of using a time-basis set can be best seen in the difference between the blue curve and the orange/red curves in (D). Artifacts introduced due to the time-basis set are highlighted with arrows and can be seen best in (E) and (F). Note that in the case of realistic EEG data, the signal is typically smooth, meaning that ripples like in (E) rarely occur. (G) The impulse response spectrum of the different smoothers. Clearly, the Fourier-set filters better than the splines, but splines allow for a sparser description of the data and could benefit in the fitting stage.

The *unfold* toolbox includes a function to remove artifactual intervals from the design matrix before fitting the model. In addition, we offer basic functionality, adapted from ERPLAB (*Lopez-Calderon & Luck, 2014*), to detect artifacts in the continuous EEG.

## Multiple solvers: LSMR & *glmnet*

Solving for the betas is not always easy in such a large model. We offer several algorithms to solve for the parameters. The currently recommended one is LSMR (*Fong & Saunders, 2011*), an iterative algorithm for sparse least-squares problems. This algorithm allows to use very large design matrices as long as they are sparse (i.e., contain mostly zeroes) which is usually the case if one uses time-expansion based on stick-functions (cf. Figs. 2 and 3A).

However, especially with data containing a high level of noise, the tradeoff between bias and variance (i.e., between under- and overfitting) might be suboptimal, meaning that the parameters estimated from the data might be only weakly predictive for held-out data (i.e., they show a high variance and tend to overfit the data). Regularization is one way to prevent overfitting of parameter estimates. In short, regularization introduces a penalty term for high beta values, effectively finding a trade-off between overfit and out-of-sample prediction. The *unfold* toolbox allows the user to specify alternative solvers that use regularization. In particular, we include the *glmnet*-solver (*Qian et al., 2013*), which allows for ridge (L2-norm), lasso (L1, leads to sparse solutions) and elastic net regularization.

The regularization parameter is automatically estimated using cross-validation but the elastic-net parameter (deciding between L1 and L2 norm) has to be specified manually. Procedures to regularize with linear deconvolution have recently been examined and validated by *Kristensen, Rivet & Guérin-Dugué (2017a)*. Effects of regularization on noisy data are also depicted in Fig. 7, which compares deconvolution results for noisy simulated data with and without regularization. In this simulation we used strongly correlated predictors ($r = 0.85$), thereby increasing collinearity. Note that collinearity in itself is only a problem for very extreme cases; specifically, if the matrix becomes ill-conditioned, the parameter solution might be instable and the estimands will "explode." Regularization can help in this situation. As can be seen in Fig. 7, the non-regularized estimates show strong variance (Figs. 7B and 7C), whereas the regularized estimates show strong bias (Figs. 7D and 7E), that is, the estimated effects are shrunk towards zero but, simultaneously, the variance of the estimate over time is greatly reduced. At this point, it is not yet clear whether and what type of regularization should be used for the standard analysis of EEG data, but we provide different solvers in *unfold* to facilitate future work on this topic. Please also see *Kristensen, Rivet & Guérin-Dugué (2017a)* for more simulation work.

## Spatial vs. temporal deconvolution

Many researchers use source reconstruction (e.g., MNE, LORETA) or blind source separation methods (e.g., ICA) to try to isolate the signal contributions of individual neural sources. In our framework, this can be understood as performing a *spatial* deconvolution of the signal that addresses the problem of volume conduction. Nevertheless, the activity time courses of each neural source may still overlap in time, for example, due to repeated stimulus presentations. To apply the deconvolution, it does not matter whether the input time series consist of raw EEG signals, or whether they are the result of spatial filtering (e.g., beamformer: *Van Veen et al., 1997*), blind source separation (*Makeig et al., 1996*; *Delorme et al., 2012*) or some other transformation (*Cohen, 2017*). To use our toolbox with other types of time series, the data has to be simply copied into the EEG.data matrix in EEGLAB. For convenience, we also offer a flag ("`ica,`" "`true`"), which allows the researcher to directly model ICA activations (stored in `EEG.icaact`), instead of the raw EEG. While we can only speculate about this issue at this point, it seem likely that a prior spatial decomposition of the data improves the performance and interpretability of the final, spatially *and* temporally deconvolved signals (*Burwell et al., 2019*).

## Comparison to a mass univariate model (without deconvolution)

The *unfold* toolbox offers the option to compute a mass univariate regression model on the same data using the exact same model but without correction for overlap. In our experience, running this model in addition to the linear deconvolution model can be helpful to understand the impact of overlap on the results. However, with this function, *unfold* can also be used as a standalone toolbox for Mass-Univariate modeling, for the (rare) cases in which an experiment does not involve any overlapping activity (e.g., from small saccades; *Dimigen et al., 2009*).

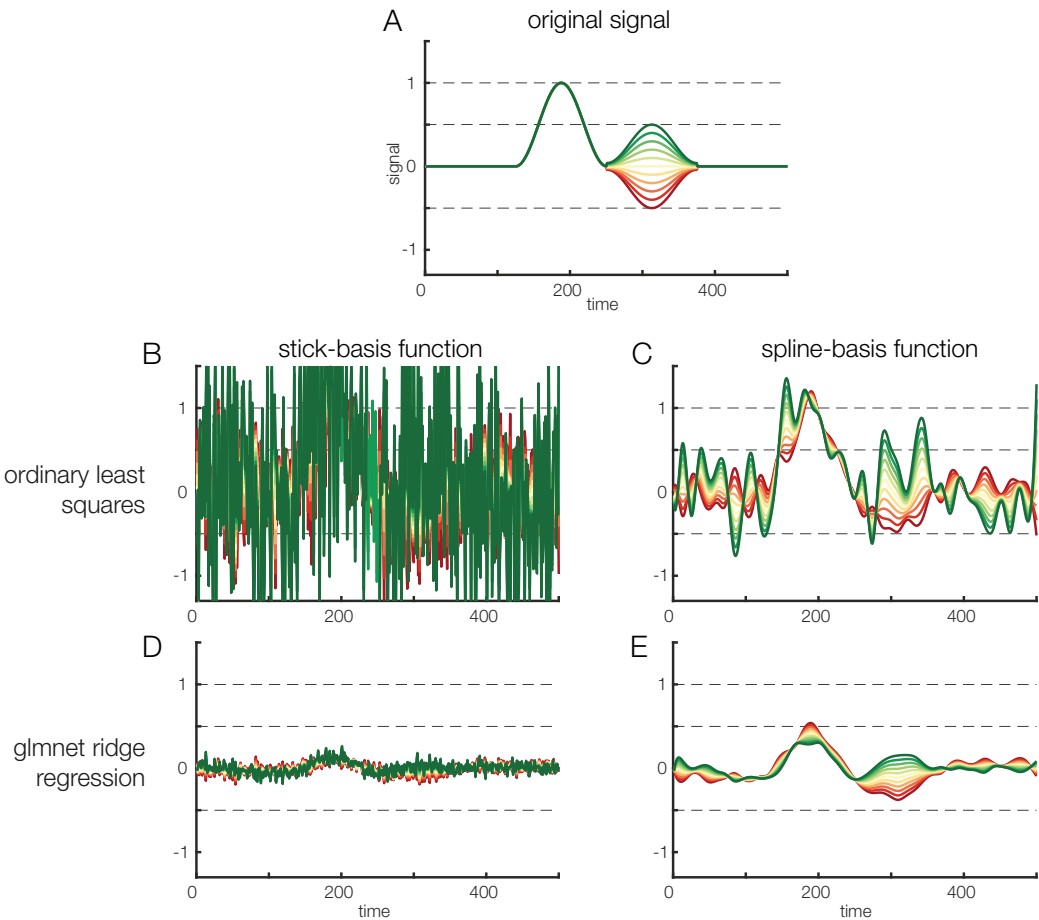

**Figure 7 Regularization options.** Effects of regularization on deconvolving noisy data. Results of regularization are shown both for a model with stick-functions and for a model with a temporal spline basis set. (A) To create an overlapped EEG signal, we convolved 38 instances of the original signal depicted in (A). The effect of a continuous covariate was randomly added to each event (see different colors in A). To make the data noisy, we added Gaussian white noise with a standard deviation of 1. Finally, to illustrate the power of regularization, we also added another random covariate to the model. This covariate had no relation to the EEG signal but was highly correlated ($r = 0.85$) to the first covariate. Thus, the model formula was: EEG $\sim$ 1 + covariate + randomCovariate. (B) Parameters recovered based on ordinary least squares regression. Due to the low signal-to-noise ratio of the data, the estimates are extremely noisy. (C) Some smoothing effect can be achieved by using time-splines as a temporal basis set instead of stick functions. (D) The same data, but deconvolved using a L2-regularized estimate (ridge regression). It is obvious that the variance of the estimate is a lot smaller. However, compared to the original signal shown in (A), the estimated signal is also much weaker, i.e., there is a strong bias. (E) L2-regularized estimates, computed with a time-spline basis set. This panel shows the usefulness of regularization: the effect structure can be recovered despite strong noise, although the recovered effect is again strongly biased (due to the variance/bias tradeoff).

## Visualization of results

*unfold* offers multiple inbuilt functions to visualize rERP results (Fig. 8). We provide functions for marginal plots over splines and continuous variables, and functions to evaluate splines/continuous covariates at specific values. For the topographical output we make use of functions from the EEGVIS toolbox (*Ehinger, 2018*).

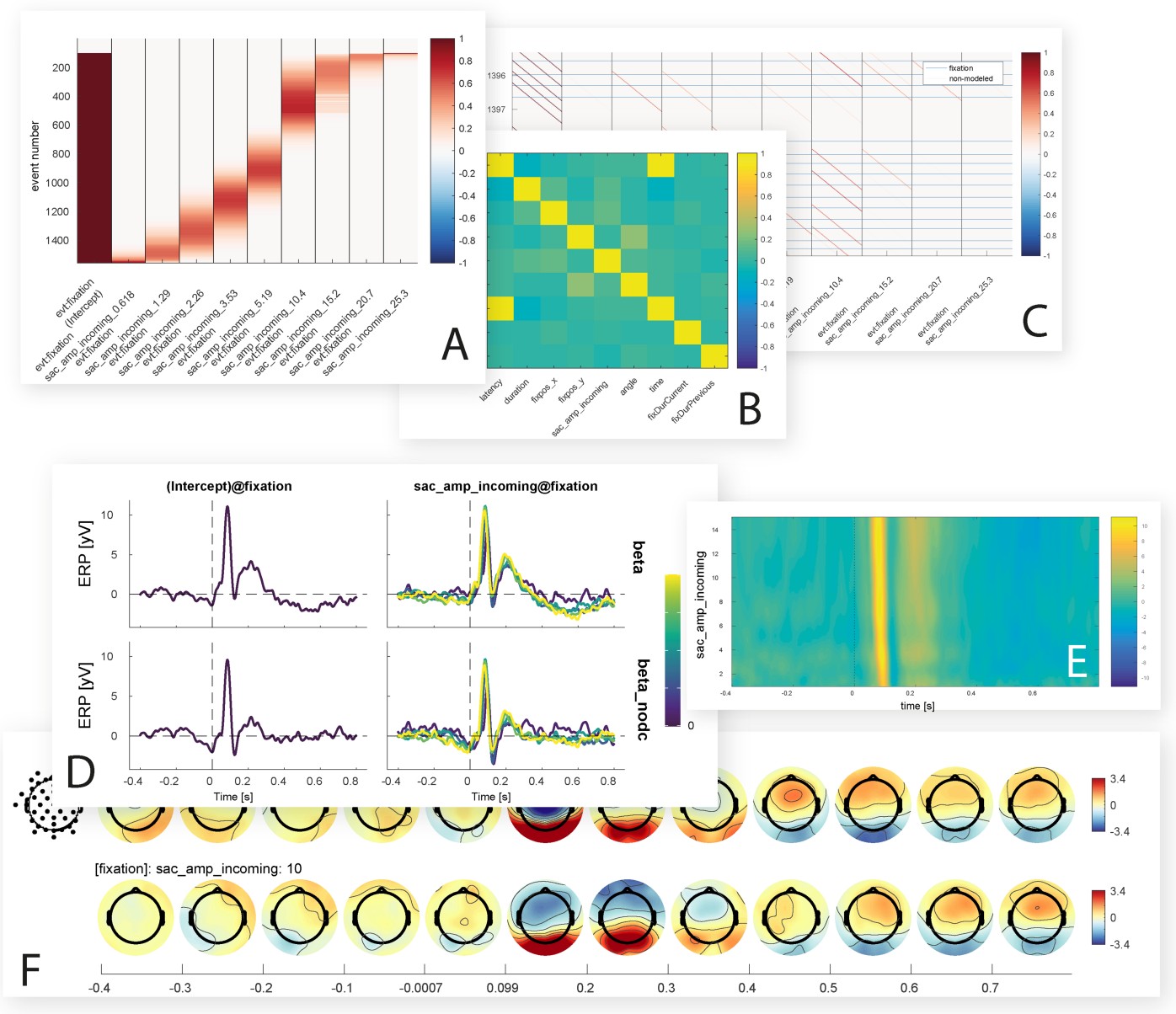

**Figure 8** **Inbuilt data visualization options.** Shown are some of the figures currently produced by the *unfold* toolbox. While setting up the model, it is possible to visualize intermediate steps of the analysis, such as the design matrix (A) covariance matrix of the predictors (B) or the time-expanded design matrix (C). After the model is computed, the beta coefficients for one or more predictors can be plotted as ERP-like waveforms with a comparison of with and without deconvolution (D), as ERP images with time against predictor value and color-coded amplitude (E), or as topographical time series (F).

## Exporting the results

*unfold* focuses on two main things: linear deconvolution and (non)-linear modeling at the single-subject level. In contrast, the toolbox itself does not offer functions for group-level statistics. However, the betas for each participant can be easily exported as plain text (.csv) or as different MATLAB structures to apply statistics with other toolboxes.

```
EEG = pop_load('eeg_example.set') % load dataset into EEGLAB

%% specify models for house/face events & button presses
cfg = []
cfg.formula    = {'y ~ 1 + cat(stim_type) + spl(luminance,5)', 'y ~ 1'}
cfg.eventtypes = {'stimulus_onset', 'button_onset'}

cfg.timelimits = [-0.5, 1]               % time window for response estimation
cfg.channel = 1:64                       % EEG channels to analyze

%% run model & plot results
run('init_unfold.m')                     % start toolbox
EEG = uf_designmat(EEG,cfg)              % create design matrix
EEG = uf_timeexpandDesignmat(EEG,cfg)   % time-expand design matrix

EEG = uf_glmfit(EEG,cfg)                 % solve regression model

ufresult = uf_condense(EEG)              % reformat results (e.g. for plotting)

uf_plotParam(ufresult,'channel',1)       % visualize rERPs (waveforms/topographies)
```

**Figure 9 A complete analysis script with *unfold*.** For further documentation and interactive tutorials visit https://www.unfoldtoolbox.org.               

A tutorial to process *unfold* results using group-level permutation tests with the TFCE-toolbox (*Mensen & Khatami, 2013*) is provided in the online documentation.

### A minimal but complete analysis script

Figure 9 shows a complete analysis script for the hypothetical face/house experiment introduced above (see Fig. 1). The complete analysis can be run with a few lines of code.

## RESULTS

In this section, we validate the *unfold* toolbox based on (1) simulated data and (2) a real dataset from a standard face recognition ERP experiment containing overlapping activities.

### Simulated data

To create simulated data, we produced overlapped data using four different response shapes, shown in the first column of Fig. 10: (1) a boxcar function, (2) a Dirac delta function, (3) a simulated auditory ERP (the same as used by *Lütkenhöner, 2010*), and (4) random pink noise. We then simulated 5 s of continuous data, during which 18 experimental events happened. Intervals between subsequent events were randomly drawn from a normal distribution ($M = 0.25$ s, SD = 0.05 s). Convolving the simulated responses with the randomly generated event latencies produced the continuous overlapped signal depicted in the third column of Fig. 10. The last column of Fig. 10 shows the

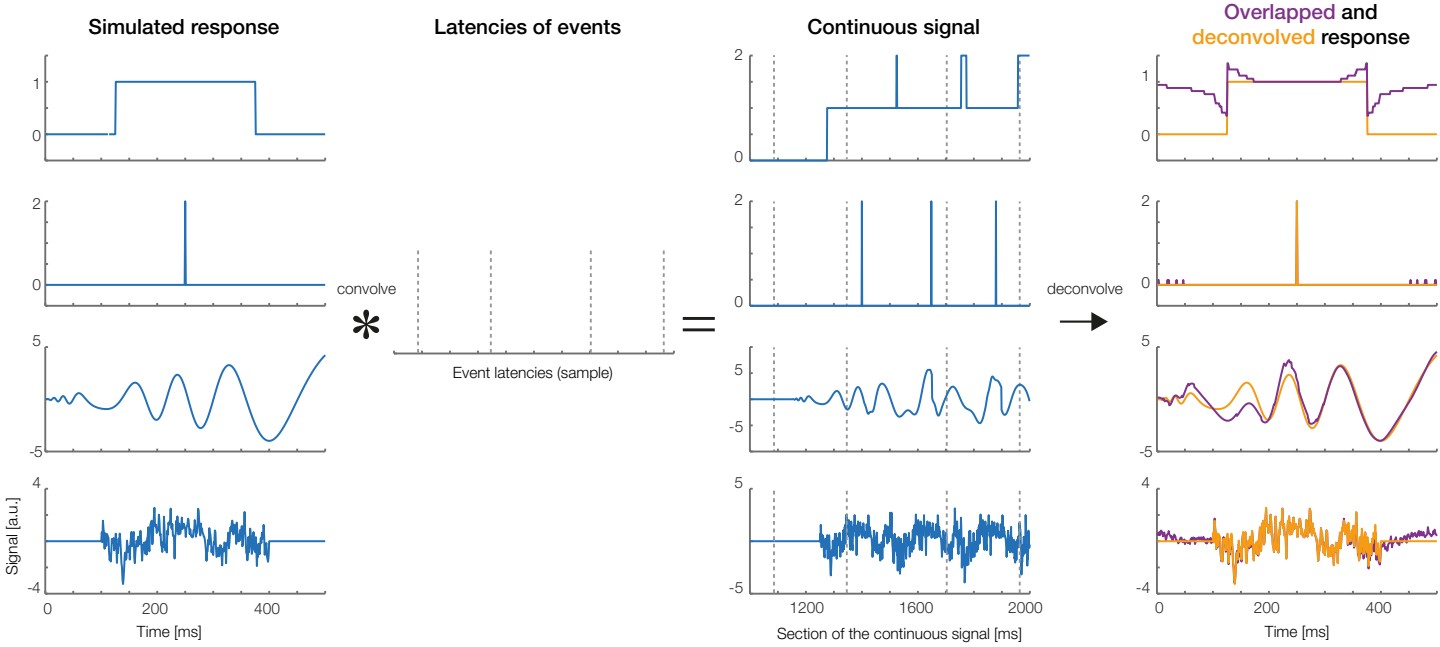

**Figure 10 Deconvolution results for simulated signals.** Four types of responses (first column: box car, Dirac function, auditory ERP, pink noise) were convolved with random event latencies (second column). A section of the resulting overlapped signal is shown in the third column. The fourth column shows the deconvolved response recovered by the *unfold* toolbox (orange lines). Overlapped responses (without deconvolution) are plotted as violet lines for comparison.

non-overlapped responses recovered by *unfold* (orange lines). For comparison, overlapped responses without deconvolution are plotted in dark red. As can be seen, *unfold* recovered the original response in all cases. The data of Fig. 1 were also simulated and then analyzed using our toolbox. Together, these simulations show that *unfold* successfully deconvolves heavily overlapping simulated signals.

## Real data example

Finally, we will also analyze a real dataset from a single participant who performed a standard ERP face discrimination experiment[1]. In this experiment, previously described in the supplementary materials of *Dimigen et al. (2009)*, participants were shown 120 different color images of human faces (7.5° × 8.5°) with a happy, angry, or neutral expression. All participants providing written informed consent before taking part in the study (at the time of data collection, there was no requirement to obtain institutional review board approval for individual ERP experiments that used standard procedures at the Department of Psychology at Humboldt University). The participants' task was to categorize the emotion of the presented face as quickly as possible using three response buttons, operated with the index, middle, and ring finger of the right hand. Each stimulus was presented for 1,350 ms. The participant's mean RT was 836 ms.

Although a central fixation cross was presented prior to each trial and participants were instructed to avoid eye movements during the task, concurrent video-based eye-tracking revealed that participants executed at least one involuntary (micro)saccades during the vast majority of trials (see also *Dimigen et al., 2009*; *Yuval-Greenberg et al., 2008*). For

[1]The same example data was also analyzed in our accompanying paper (*Dimigen & Ehinger, 2019*), but with a different focus. Further details on this dataset are given in *Dimigen et al. (2009)* or *Dimigen & Ehinger (2019)*.

the participant analyzed here, the median amplitude of these small saccades was 0.6° and most were aimed at the mouth region of the presented faces, which was most informative for the emotion discrimination task.

This means that our stimulus-locked ERPs are contaminated with two other processes: visually-evoked potentials (lambda waves) generated by the retinal image motion produced by the (micro)saccades (*Gaarder et al., 1964*; *Dimigen et al., 2009*) and motor processes related to preparing and executing the finger movement.

To disentangle these potentials with *unfold*, we specified three events: Stimulus onset, saccade onset, and button press. For this simple demonstration, we modeled both stimulus onsets and button press events using only an intercept term (y ~ 1), that is, regardless of emotion. For the saccade onsets, we included both an intercept as well as saccade amplitude as a continuous predictor, because larger saccades are followed by larger lambda waves (*Gaarder et al., 1964*; *Dimigen et al., 2009*). Because this relationship is non-linear (*Dandekar et al., 2012b*) we used a set of 5 splines in the formula, y ~ 1 + spl (saccade_amplitude,5). Brain responses were modeled in the time window from −1.5 to 1 s around each event. Before fitting the model, we removed all intervals from the design matrix in which the recorded activity at any channels differed by >250 µV within a window of 2 s.

Figure 11 presents the results for occipital electrode Oz and the signal both with (in red) and without (blue) the modeling and removal of overlapping activity. The large effect of overlapping activity can be clearly seen in the averaged ERP waveforms (top row in Figs. 11C–11E). In the corresponding panels below that, we see the color-coded single trial activity (*erpimages*), in which segments time-locked to one type of event (e.g., stimulus onset) were sorted by the latency of the temporally adjacent event (e.g., saccade onset). These panels clearly show the overlapping activity and how it was successfully removed by the deconvolution. In particular, we wish to highlight the substantial effect of overlap correction on the shape of both the stimulus-onset ERP (elicited by the faces) and the response-related ERP (elicited by the button press), despite the fact that average RT was relatively long (>800 ms) in this task. Microsaccades have an additional distorting effect (*Dimigen et al., 2009*). We can therefore easily imagine how without any overlap correction, differences in mean RT and microsaccade occurrence between conditions will create spurious condition effects in the stimulus-ERP. A more complex application where we correct for similar spurious effects in a natural reading EEG experiment with 48 participants is found in *Dimigen & Ehinger (2019)*. The data and code to reproduce Fig. 11 can be found at https://osf.io/wbz7x/.

## DISCUSSION

Human behavior in natural environments is characterized by complex motor actions and quasi-continuous, multisensory stimulation. Brain signals recorded under such conditions are characterized by overlapping activity evoked by different processes and typically also influenced by a host of confounding variables that are difficult or impossible to orthogonalize under quasi-experimental conditions. However, even in traditional, highly controlled laboratory experiments, it is often unrealistic to match all stimulus properties

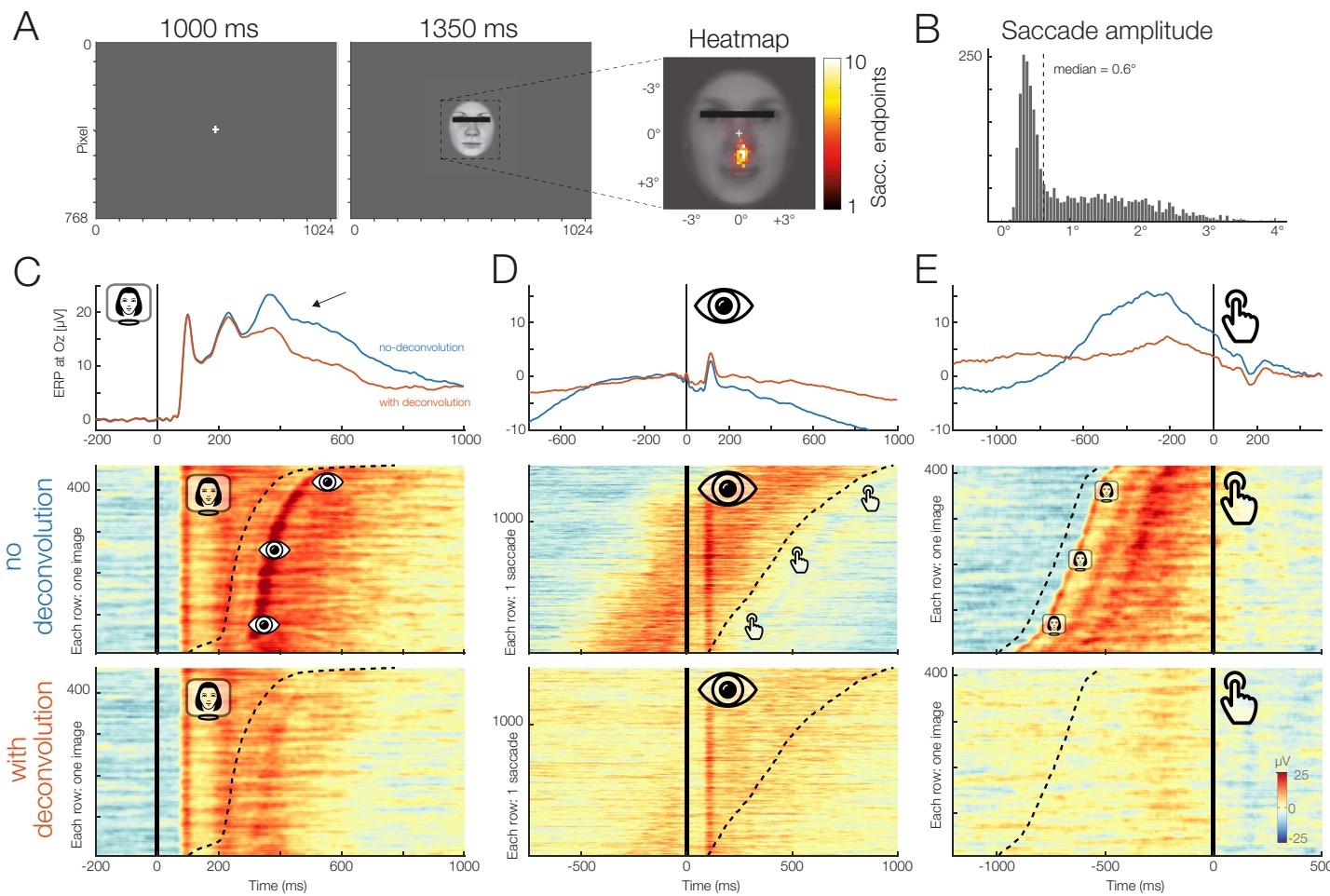

**Figure 11 Example dataset with stimulus onsets, eye movements, and button presses.** (A) Panel adapted from *Dimigen & Ehinger (2019)*. The participant was shown a stimulus for 1,350 ms. (B) The subject was instructed to keep fixation, but as the heatmap shows, made many small involuntary saccades towards the mouth region of the presented stimuli. Each saccade also elicits a visually-evoked response (lambda waves). (C–E) Latency-sorted and color-coded single-trial potentials at electrode Oz over visual cortex (second row) reveal that the vast majority of trials contain not only the neural response to the face (C) but also hidden visual potentials evoked by involuntary microsaccades (D) as well as motor potentials from preparing the button press (E). Deconvolution modeling with *unfold* allows us to isolate and remove these different signal contributions (see "no deconvolution" vs. "with deconvolution"), resulting in corrected ERP waveforms for each process (blue vs. red waveforms). This reveals, for example, that a significant part of the P300 evoked by faces (arrow in (C)) is really due to microsaccades and button presses and not the stimulus presentation.

between conditions, in particular if the stimuli are high-dimensional, such as words (e.g., word length, lexical frequency, orthographic neighborhood size, semantic richness, number of meanings, etc.) or faces (e.g., luminance, contrast, power spectrum, size, gender, age, facial expression, familiarity, etc.). In addition, as we demonstrate here, even simple EEG experiments often contain overlapping neural responses from multiple different processes such as stimulus onsets, eye movements, or button presses. Deconvolution modeling allows us to disentangle and isolate these different influences to improve our understanding of the data.

In this article, we presented *unfold*, which deconvolves overlapping potentials and controls for linear or non-linear influences of covariates on the EEG. In the following,

we will discuss in more detail the assumptions, possibilities, and existing limitations of this approach as well as current and future applications.

## Where can linear deconvolution be applied?

Linear deconvolution can be applied to many types of paradigms and data. As shown above, one application is to separate stimulus- and response-related components in traditional ERP studies (see also *Ouyang et al., 2011*; *Ouyang, Sommer & Zhou, 2015*). Deconvolution is also particularly useful with complex ERP designs that involve, for example, multimodal streams of visual, tactile, and auditory stimuli (*Spitzer, Blankenburg & Summerfield, 2016*). Deconvolution is also helpful in paradigms where it is problematic to find a neutral interval to place a baseline, for example, in experiments with fast tone sequences (*Lütkenhöner, 2010*). In ERP research, the interval for baseline correction is usually placed immediately before stimulus onset, but activity in this interval can vary systematically between conditions due to overlapping activity, for example, in self-paced paradigms (*Ditman, Holcomb & Kuperberg, 2007*). This problem can be solved by first deconvolving the signal and then applying the baseline subtraction to the resulting isolated responses.

## Time-continuous covariates

It is also possible to add time-continuous signals as predictors to the design matrix (*Lalor et al., 2006*; *Crosse et al., 2016*). Examples for continuous signals that could be added as predictors include the luminance profile of a continuously flickering stimulus (*Lalor et al., 2006*; *VanRullen & MacDonald, 2012*), the sound envelope of an audio or speech signal (with temporal lags to model the auditory temporal response function *Crosse et al., 2016*), the participants' gaze position or pupil size (from concurrent eye-tracking *Dimigen & Ehinger, 2019*), but also more abstract time series, such as predictions from a cognitive computational model. Including time-continuous covariates such as gait-signals, movement features, or environmental sounds could also improve the model fit in mobile EEG situations (*Ehinger et al., 2014*; *Gramann et al., 2014*). The resulting betas of time-continuous covariates are often named (multivariate) temporal response functions (mTRFs, but not consistently so, e.g., *Broderick et al., 2018* call FIR-deconvolved betas TRFs). We propose to follow this nomenclature to distinguish it from (stick-function-fitted) rERPs as we have been using in this paper. Of course, one could combine both approaches and fit mTRFs and rERPs simultaneously.

## Underlying assumptions

A fundamental assumption of traditional ERP averaging is that the shape of the underlying neural response is identical in all trials belonging to the same condition. Trials with short and long manual RTs are therefore usually averaged together. Similarly, with linear deconvolution modeling, we assume that the brain response is the same for all events of a given type. However, like in traditional ERP analyses, we also assume that the neural response is independent of the interval between two subsequent events (e.g., the interval

between a stimulus and a manual response). This is probably a simplification, since neural activity will likely differ between trials with a slow or fast reaction.

A related assumption concerns sequences of events: processing one stimulus can change the processing of a following stimulus, for instance due to adaptation, priming, or attentional effects. We want to note that if such sequential effects occur often enough in an experiment, they can be explicitly modeled; for example, on could add an additional predictors coding whether a stimulus is repeated or not or whether it occurred early or late in a sequence of stimuli. We hope that the *unfold* toolbox will facilitate the analysis of simulations on these issues and also propose to analyze experiments where temporal overlap is experimentally varied.

## Modeling non-linear effects

Non-linear predictors can have considerable advantages over linear predictors. However, one issue that is currently unresolved is how to select an appropriate number of spline functions to model a non-linear effect without under- or overfitting the data. While automatic selection methods exist (e.g., based on generalized cross-validation, *Wood, 2017*), their high computational cost of repeatedly deconvolving the data precluded us from using these techniques. In the current implementation of *unfold*, we assume the same number of splines are needed for all parts of the response. But it is possible, for example, that with a constant number of splines the baseline interval is overfitted, whereas the true response is underfitted. Therefore, algorithms to find smoothing parameters need to take into account that the amount of smoothing changes throughout the response. Choosing the correct number of splines that neither overfit nor underfit the data is an important question to resolve, and again, we hope that the *unfold* toolbox will facilitate future simulation studies, new algorithms, and new experiments on this issue.

## Time-frequency analysis

While all example analyses presented here were conducted in the time domain, it is also possible to model and deconvolve overlapping time-frequency representations with *unfold* (see also *Litvak et al., 2013*). One simple option is to enter the band-bass filtered EEG signal into the model (or alternatively bandpass filter the estimated betas, but be aware of boundary filter artefacts). This would model the evoked potentials. One could also estimate the instantaneous power for induced potentials. For instance *Ossandón, König & Heed (2019)* deconvolved the instantaneous power after band pass filtering the alpha band. But this assumes that the overlap is linear in power. *Litvak et al. (2013)* used an empirical approach based on model fit to decide between three possible power transformations (power, sqrt(power), and log(power)) and found slightly better model fit for the sqrt(power) transform. Alternatively, complex linear regression could be a solution (*Hussin, Abdullah & Mohamed, 2010*), but future work is needed here and we recommend more simulation studies prior to such time-frequency work.

## Choice of modeling parameters

In a traditional ERP analysis, the researcher has to set numerous analysis parameters, which will influence the final ERP results (e.g., filter settings, epoch length, baseline interval). Similarly, with deconvolution we have to make these and additional choices. Because the deconvolution approach is still in its infancy, there are not yet clearly established best-practices for all of the necessary settings. However, in the following, we discuss some basic and advanced parameters:

Basic settings/options

1) *Sampling rate.* Deconvolution can be applied to data recorded at any sampling rate, but some researchers have temporally downsampled their data (e.g., to 100 Hz; *Sassenhagen, 2018*) before deconvolution. Aside from the obvious loss of temporal resolution that results from downsampling, we (anecdotally) have not observed a benefit for the stability of the estimation; in our experience, downsampling made the fitting process faster but not necessarily better.

2) *Epoch size.* The time limits for the deconvolution should be chosen so that the entire event-related response is modeled. For motor responses, such as saccades or button presses, it is therefore necessary to also include a sufficient number of timepoints before the event in order to capture (pre)motor potentials.

3) *Baseline correction.* Baseline correction can be applied directly to the resulting betas (*Smith & Kutas, 2015b*). Whether baseline correction is always a good idea is still up to debate and in future work, it might be possible to include the baseline voltages directly into the GLM (*Alday, 2019*).

Advanced settings/options

4) *Number of splines.* The number of splines for a non-linear predictor is a difficult parameter to set, because it clearly depends on the underlying relation between predictor and ERP. Usually the number of splines is determined by penalized least squares, which is not supported by *unfold*. In our application to combined EEG and eye-tracking experiments (*Dimigen & Ehinger, 2019*) we found that EEG effects like that of saccade size can be modeled by about 5 splines, whereas a much higher number of splines (e.g., 10) clearly overfitted the single-subject data. In future implementations it might be possible to estimate the splines using penalized regression, mixed model fitting (*Wood, 2017*) or using cross-validation.

5) *Placement of splines.* The splines have to be placed along the range of the predictor variable. In *unfold*, the default placement of splines is on the ($N_{splines}$-2) quantiles of the predictor. Other placements are possible, and users can specify their own placements if necessary (*Harrell, 2015*, p. 26).

6) *Regularization.* Whether to use regularization at all and if so, what type of regularization to use, is currently not established. We recommend *Kristensen, Rivet & Guérin-Dugué (2017a)* as a starting point.

7) *Time basis used to generate $X_{dc}$.* As described above, it is also possible to use a set of *temporal* basis functions (e.g., a Fourier set or set of temporal splines) to reduce

the number of columns added to the design matrix in the time-expansion step. While *unfold* already implements this option, we follow for now the recommendation of *Smith & Kutas (2015b)* that simple stick functions (FIR) should be used for most applications.

## Outlook: integration with linear mixed models

In recent years, linear mixed-effects models (LMM, e.g., *Gelman & Hill, 2007*) have been slowly superseding traditional models like repeated-measures ANOVA or the two-stage hierarchical approach used here. LMMs allow to model the hierarchical structure of single-subject and group-level data directly and have several other advantages, for example, when analyzing unbalanced designs (*Baayen, Davidson & Bates, 2007*; *Kliegl et al., 2010*). Combining LMMs with linear deconvolution is theoretically possible (*Ehinger, 2019*). The main challenge is that one needs to fit all continuous EEG datasets of all participants at the same time. Thus, currently, the high computational cost of fitting such large models precludes us from taking advantage of mixed models. Nevertheless, recent progress with similarly large models (*Wood et al., 2017*) shows that the combination of LMMs with deconvolution modeling might be computationally feasible in future implementations.

## Other physiological signals

Finally, it is also possible to model other types of overlapping psychophysiological signals with *unfold*, such as overlapping magnetic fields (MEG, *Litvak et al., 2013*), pupil dilations (*Wierda et al., 2012*; *Gagl, Hawelka & Hutzler, 2011*) or skin conductance responses (*Bach et al., 2009*).

## CONCLUSIONS

In summary, *unfold* offers an integrated environment to analyze psychophysiological data influenced by overlapping responses, (non)linear covariates, or both. As we show above, this analysis strategy can be beneficial even in the case of "traditional," highly-controlled ERP experiments. It also allows us to record EEG data under more natural situations, for example, those with unconstrained eye movement behavior, which is typical for the emerging fields of virtual reality and mobile brain/body imaging studies. Applications of *unfold* to free viewing studies can be found in an accompanying paper (*Dimigen & Ehinger, 2019*). The toolbox is freely available at http://www.unfoldtoolbox.org with tutorials and documentation.

### Funding

The project was supported by the European Commission Horizon (H2020-FETPROACT-2014 364 641321-socSMCs). The collection of the face dataset was supported by DFG Research Group 868, project A2. The funders had no role in the study design, data collection and analysis, decision to publish, or preparation of the manuscript.

### Grant Disclosures

The following grant information was disclosed by the authors:

European Commission Horizon: H2020-FETPROACT-2014 364 641321-socSMCs.

DFG Research Group 868, project A2.

### Competing Interests

The authors declare that they have no competing interests.

### Author Contributions

- Benedikt V. Ehinger analyzed the data, contributed reagents/materials/analysis tools, prepared figures and/or tables, authored or reviewed drafts of the paper, approved the final draft, wrote the toolbox.
- Olaf Dimigen conceived and designed the experiments, performed the experiments, analyzed the data, contributed reagents/materials/analysis tools, prepared figures and/or tables, authored or reviewed drafts of the paper, approved the final draft, co-developed the toolbox.

### Human Ethics

The following information was supplied relating to ethical approvals (i.e., approving body and any reference numbers):

For this methods paper, we reanalyzed data of a single subject from an ERP dataset that was previously published elsewhere (in: *Dimigen et al., 2009*, Journal of Neuroscience, Supplementary Materials). These data were originally recorded in 2008 and therefore at a time before it was required to apply for a study-specific ethics commission approval to run standard ERP experiments at the Humboldt-University at Berlin.

### Data Availability

Data is available at the Open Science Foundation (https://osf.io/wbz7x) and Github (https://github.com/unfoldtoolbox/unfold).

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
