# Peer review of "Unfold: an integrated toolbox for overlap correction, non-linear modeling, and regression-based EEG analysis"

_PeerJ, doi:10.7717/peerj.7838_

## Round 0.1 · original submission · Minor Revisions

Dear Author, We would like your manuscript to undergo minor revisions as per comments of the peer reviewers.

Thank you.

·

Basic reporting

I have openly peer-reviewed an earlier version of this manuscript.
The authors have addressed my remarks, and these are reported in the current paper.
I thus endorse its publication in its present form.

Experimental design

no comment

Validity of the findings

no comment

Reviewer 2 ·

Basic reporting

This manuscript presented a toolbox for a data analysis framework that is more versatile than conventional trial-averaging ERP approach in the sense that heterogeneous overlapping events and complicated effects can be modelled in the framework. Thus, more sophisticated EEG experiments are encouraged, providing the help of this unfold toolbox in the subsequent data analyses.
The author’s endeavour underlying this work is highly appreciated, as well as its potential contribution to future scientific investigations. As mentioned by the author, several tools that have been developed with similar purpose are either difficult to access/use, or not being maintained/further developed. This somehow reflects the challenge in this area: sophistication in the methodologies that are oriented to dealing with complex, closer-to-reality paradigms requires exponentially increasing amount of development and maintenance effort, which hinders its development and user application. The authors have made a great effort in constructing, validating, and facilitating this new data analysis framework of EEG data analysis under complex designs.
I recommend the publication of this work without much reservation. But I do hope my following comments can be addressed. I list all my comments without distinguishing them as major or minor.
- Figure 2: ‘response to A at t=1’, ‘response to A at t=5’, ‘response to B at t=4’, should ‘t’s here be ‘tau’ (the greek letter)?
- line 193: should the subscript ‘t’ after EEG be ‘25’, according to Figure 2? ‘t’ refers to any time point, but not every time point of EEG can be expressed as the right part in line 193, which is a specific case. This also applies to line 194 ‘at time point t’. It should be ‘at time point t = 25’.
- Why is the design matrix named Xdc? What does ‘dc’ stand for?
- line 206, 207: How Xdc is constructed is actually quite essential. I would not suggest to remind readers to skip this part.
- line 259: this comes at the cost of very large (size…). The left parenthesis should pass the word ‘size’.
- line 85, why is Hanke & Halchenko, 2011 cited here?
- line 135, ‘have existing’ -> ‘have existed’
- line 312, 313: “A benefit of using temporal basis functions rather than the simple stick functions is that less unknown parameters need to be estimated.” Is this true if, for example, a full set of Fourier basis is used? In Figure 3 it seems that only a limited frequency band of Fourier basis is used. This is equivalent to low-pass filtering. Thus the reduction of parameter to be estimated is at the sacrifice of losing information. Similar thing can be done by using stick function in down-sampled data.
- line 393, 394: it is particularly necessary to split the model specification into two parts? This is somewhat counter-intuitive, because the second line seems to have overwrite the first line. And since stimulus events and response events do not differentiate each other in the linear model (they are both regressors), it would look more sensible to put them in the same formulation.
- line 429 to 431: the comparison in Figure 6 is somewhat anecdotal. It totally depends on the parameters setting of the basis functions. The major point being delivered here seem to be the filtering effect. The reason why and in what cases basis functions should be considered should be more conceptually explained. For example, one advantage is the computational speed.
- line 445: “The way to handle artifacts in a linear deconvolution model is therefore to detect – but not to remove – the intervals containing artifacts in the continuous dataset”. What is the difference between removing the data and blanking out the data?
- The introduction of regularization may not be very clear to readers. It would be great to concisely introduce the rational/benefit of applying regularization to readers before introducing the implementation. It is stated that “The regularization parameter is automatically estimated using cross-validation.”, but then later “it is not yet clear whether and how much regularization should be used for the standard analysis of EEG data”. Since it can be automatically estimated, why should it be concerned about “how much regularization should be used”? Plus, the results in Figure 7C may be further improved by low-pass filtered (using less splines), which may produce less noisy and also less biased result.

Experimental design

This paper is a toolbox paper, thus commenting on experimental design is not applicable.

Validity of the findings

Again, this paper is a toolbox paper, thus commenting on validity of findings is not applicable.

Reviewer 3 ·

Basic reporting

The manuscript is very clearly written, the figures are relevant and high quality. The structure is clear and it was a pleasure to read.

I am not sure of the PeerJ policy but many journals have a policy where related manuscripts in preparation or under consideration elsewhere should be explicitly highlighted and often provided with submission. Here there seems to be a lot of overlap (to the extent that many of the figures seem to be common between the two papers) with another paper by the same authors cited as “In Preparation”. I would like to flag this for editorial consideration.

I miss a clear statement regarding raw data availability and location (for the EEG data used for Figure 10). Based on PeerJ policy I would expect to see these posted to a suitable repository before publication. An earlier paper is cited (Dimigen et al. 2009) but does not have a data availability statement. The other paper cited for the data is referenced as in prep (see above).

Experimental design

The manuscript describes a software tool, but I would say it goes beyond that by providing an accessible introduction to a broad modelling approach which is well pitched to a target audience of practising neuroimagers. The approach and software proposed is novel and should prove very useful to the field.

Validity of the findings

I tried to run the software myself I did seem to run into some errors during the tests. I am using matlab r2018b on a mac with EEGLAB 14_1_2b:

The link from the repo readme to https://www.unfoldtoolbox.org/docs_sphinx/_build/html/toolboxtutorials.html seems to be broken (404).

The tutorials I tried seemed to run OK but it would be nice to have them in Matlab scripts as well as the html to avoid having to repeatedly copy and paste (perhaps I missed something here?) They look like they have been generated as Matlab “Live Scripts” – in which case having the source script to manually run each cell would be nice.

Get an error running the tests:
Modeling 234 event(s) of [stimulusA] using formula: y~1+continuousA
Error using uf_continuousArtifactDetect (line 74)
mode error: undefined argument 'combineSegements'

Error in test_continuousArtifact (line 17)
winrej = uf_continuousArtifactDetect(EEG,cfgClean);

Error in uf_tests (line 10)
test_continuousArtifact

Fix: typo line 10 of test_continuousArtifact

The next error I got was:
uf_timeexpandDesignmat(): Timeexpanding the designmatrix...
...done
Portions of data removed split up by each event in EEG.event
Type: stimulusA modelled eventtime: 596.80s rejected eventtime: 3.80s percent removed: 0.6%

Removing 0.65% of rows from design matrix (filling them with zeros)
eeg_checkset note: upper time limit (xmax) adjusted so (xmax-xmin)*srate+1 = number of frames
Event resorted by increasing latencies.
Event resorted by increasing latencies.
Multiple events with separate model-formula detected
Modeling 139 event(s) of [stimulus1] using formula: y~1
Modeling 138 event(s) of [stimulus2] using formula: y~1+conditionA*continuousA
Modeling 138 event(s) of [stimulus3] using formula: y~1+continuousA
Error using histc
Edge vector must be monotonically non-decreasing.

Error in Bernstein (line 125)
[trash col] = histc(x,tt); %#ok

Error in default_spline (line 11)
a = Bernstein(paramValues',knots,[],4,[],0);

Error in uf_designmat_spline (line 169)
Xspline = spl.splinefunction(spl.paramValues,spl.knots);

Error in uf_designmat (line 567)
[EEG, ~,nanlist] =
uf_designmat_spline(EEG,'name',cfg.spline{s}{1},'nsplines',cfg.spline{s}{2}(1),'paramValues',t{:,cfg.spline{s}{1}},'splinespacing',cfg.splinespacing,'splinefunction',cfg.spline{s}{3},'cyclical_bounds',bounds);

Error in uf_designmat (line 150)
EEG2 = uf_designmat(EEG,cfgSingle);

Error in test_designmat (line 19)
uf_designmat(EEGsim,cfgDesign);

Error in uf_tests (line 11)
test_designmat

125 [trash col] = histc(x,tt); %#ok

Additional comments

The manuscript describes a MATLAB toolbox for linear regression modelling of EEG data. It seems to be a flexible and powerful tool, with several useful features that are not trivial for individual labs to implement. It particularly focusses on the ability to distinguish temporally overlapping events through linear modelling with time shifted regressors (I believe FIR basis in fMRI literature).

I have one two major comments for small additions (maybe a short paragraph could be added for each point).

1. I understand from use of GLM in fMRI that a major issue when developing and applying complex linear models in practise is the issue of co-linearity of predictors – which I understand can cause problems for model fitting and interpretation of the obtained betas. As far as I can see this is not mentioned at all in the current manuscript. I think it would be good to at least discuss this issue, perhaps with reference to the extensive fMRI literature on the subject. In fMRI the design matrix is often orthogonalized. Is that something Unfold supports or would make sense for EEG? Could you highlight perhaps where the examples you consider might result in collinearity that could cause problems (taking into account the time expansion approaches etc.). The example in Figure 7 seems to have highly correlated predictors which I would have thought could be problematic but there is no comment on that here. Is it less of a problem for EEG than fMRI? I believe SPM has diagnostic tools to highlight when a design matrix has correlated predictors. Does Unfold have any such functionality?

2. I would also like to see maybe more practical discussion about the interplay between sampling rate and time expansion. The examples give 5 time expansion points, but what is the sampling rate considered here. Presumably at 1000Hz vs 100Hz sampling the behaviour is very different. Could the authors comment a bit somewhere on what sort of parameters they might consider optimal for traditional EEG designs? What number of trials are broadly necessary for say an ERP with 5 time points considered like here, what about if you considered more (say 50 time points at a higher sampling rate) – would more trials be required? Would co-linearity be more of a problem? Presumably downsampling might increase statistical power somewhat, but reduce sensitivity to latency differences? Just some broad strokes comments on things to be aware of and perhaps some practical advice would be useful. Similarly, what would the difference between low pass filtering the signal first, and filtering in the model. So there are two issues, one is sampling rate and the other is filtering. I.e. could filter before to <30Hz and keep at 1000Hz. Or could filter to <30Hz and downsample to 100Hz. What are the implications for the temporal expansion?

Apart from those suggestions I have only minor comments:

In the introduction I would like to see a paragraph that discusses more broadly alternative approaches to the problem. I am thinking for example to mention source separation algorithms, both statistical (such as generalised eigenvalue decomposition, see e.g. [1], or ICA see e.g. [2,3]), and spatial (e.g. source localisation such as LORETA etc.). Of course the regression approach can still be applied to such data, but it could be worth noting that these have traditionally been use to address some of the issues (separation of motor response signal from visual evoked signal for example, even when they might be overlapping in time). There are also alternative statistical approaches such as information theory [4] or MVPA (many possible citations). But its fair to say that none of these can address the temporal deconvolution of effects in the same statistical or spatial source. Along these lines it might also be worth noting in the discussion that this temporal deconvolution might fruitfully be combined with other statistical approaches (such as MVPA) in the same way it is with fMRI (where MVPA applies on betas from a linear model to deconvolve overlapping responses).

[1] http://www.sciencedirect.com/science/article/pii/S0165027016303004
[2] https://journals.plos.org/plosone/article?id=10.1371/journal.pone.0030135
[3] http://papers.nips.cc/paper/1091-independent-component-analysis-of-electroencephalographic-data.pdf
[4] http://onlinelibrary.wiley.com/doi/10.1002/hbm.23471/abstract

Line 93: “threshold-free cluster-based permutation”. My understanding is that TFCE is a cluster enhancement, but that inference is still performed test-wise rather than cluster-wise (which is a big advantage for interpreting results in terms of localisation). So I would maybe rephrase this: “… integrates permutation based maximum statistics with threshold-free cluster enhancement (…)” (or if not maximum statistics approach describe in more detail the permutation approach used). Key point is don’t say cluster-based if its TFCE. Same comment L513. Not cluster-based inference.

Line 119: Pedantic comment, not required to address. To my mind systematically different RTs between classes of stimuli must reflect a genuine difference in the way the brain processes those stimuli – so its not clear to me why they would be labelled spurious.

Line 181: mu is the mean value, y is the EEG signal? Found this confusingly phrased. The second presentation seems clearer (line 257).

Line 278: could introduce earlier what a spline is – ie just that it is a local smooth function, maybe say the form of the ones used. Also knots are referenced but not defined (L418). Maybe could add a sentence or two briefly defining splines and knots with a reference, or expand a bit. Line 420, “knot placement is on the quantiles of the predictor” – what quantiles? (quartiles, percentiles?)

Line 292: related to the point about practical advice re sampling rate. “determine the number of splines prior to the analysis” – how? At least give some idea of the approach or factors or a citation or something.

Fig 4 caption. Typo inbuild -> inbuilt

Line 518: Why is such a large prestimulus window used (500ms). In general any practical guidance on how to set this? (see major comment about practical advice on parameters / sampling rate etc.)

I think the authors could add to the discussion (or introduction) perhaps a comment that often extracting single trial measures of ERP are interesting (e.g. peak, latency). See for example [5]. Could they comment on whether Unfold could be used to address this sort of question (ie extracting different ERP peaks or latencies on individual trials, either from single trial beta modelling as used in fMRI or some other way)?

[5] http://jn.physiology.org/content/106/6/3216

Reviewer 4 ·

Basic reporting

No comment

Experimental design

No comment

Validity of the findings

No comment

Additional comments

I've written no comment in all of the above fields, simply because I can see no respects in which the article fails to meet the required standard. This is an excellent article; well-written, thorough and clear. It clearly identifies the problem, provides appropriate theoretical background, and clearly explains how their work differs from existing approaches. I found that any questions which came to mind while reading (for example, how or if it would be possible to extend the approach to dealing with data from multiple participants) had already been considered by the authors and were answered as I read further. The approach taken for the toolbox is exciting and innovative; the documentation is thorough and detailed, with clear examples of the workflow and usage of the toolbox. The potential applications and limitations of the approach are also clearly considered. The authors have made every effort to make their work accessible and useful, providing access to raw data and scripts etc. I think it makes an excellent contribution to the field as is.

I have one or two minor typos to point out:

The word "inbuild" appears several times - e.g. Captions of figure 4 and figure 8. This should be "inbuilt".
Line 357 should read "We begin the modelling process by..."
Line 572 sentence is unfinished.

---

## Round 0.2 · accepted · Accept

Dear Authors,Congratulations,Your manuscript has been accepted for publication in PeerJ.

Thank you.

Reviewer 2 ·

Basic reporting

no comment

Experimental design

no comment

Validity of the findings

no comment

Additional comments

my previous comments were well addressed.

Reviewer 3 ·

Basic reporting

no comment

Experimental design

no comment

Validity of the findings

no comment

Additional comments

The authors have very thoroughly responded to and addressed all my comments.

I would congratulate them on a very nice paper that described what I'm sure will be a very useful and influential tool.


Minor note:
Line 413: the equation shows the parametr 5 but the text refers to 10 splines. This is a bit confusing. Maybe text needs to be changed to 5 to match?